# 6DGS: Enhanced Direction-Aware Gaussian Splatting for Volumetric Rendering

**Zhongpai Gao,**\* **Benjamin Planche, Meng Zheng, Anwesa Choudhuri, Terrence Chen, Ziyan Wu**
United Imaging Intelligence, Boston MA, USA
`{first.last}@uii-ai.com`

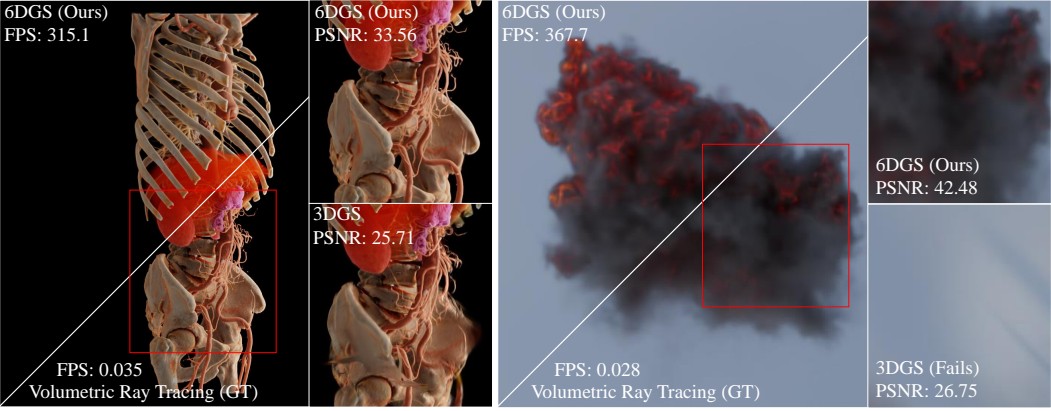

Figure 1: Visualizations of volumetric rendering. Top-left: our 6DGS rendering; bottom-right: physically-based rendering using ray/path tracing; right: comparison with 3DGS over the red regions.

## ABSTRACT

Novel view synthesis has advanced significantly with the development of neural radiance fields (NeRF) and 3D Gaussian splatting (3DGS). However, achieving high quality without compromising real-time rendering remains challenging, particularly for physically-based rendering using ray/path tracing with view-dependent effects. Recently, N-dimensional Gaussians (N-DG) introduced a 6D spatial-angular representation to better incorporate view-dependent effects, but the Gaussian representation and control scheme are sub-optimal. In this paper, we revisit 6D Gaussians and introduce 6D Gaussian Splatting (6DGS), which enhances color and opacity representations and leverages the additional directional information in the 6D space for optimized Gaussian control. Our approach is fully compatible with the 3DGS framework and significantly improves real-time radiance field rendering by better modeling view-dependent effects and fine details. Experiments demonstrate that 6DGS significantly outperforms 3DGS and N-DG, achieving up to a 15.73 dB improvement in PSNR with a reduction of 66.5% Gaussian points compared to 3DGS. The project page is: `https://gaozhongpai.github.io/6dgs/`.

## 1 INTRODUCTION

Novel view synthesis enables the generation of new viewpoints of a scene from limited images, underpinning applications in virtual reality, augmented reality, and realistic rendering for films and games. A significant milestone was the development of neural radiance fields (NeRF) (Mildenhall et al., 2020), representing scenes as continuous volumetric functions that map 3D coordinates and viewing directions to color and density values. While NeRF captures intricate details and complex lighting effects, its reliance on computationally intensive neural networks makes real-time rendering challenging. To address this, 3D Gaussian splatting (3DGS) (Kerbl et al., 2023) was introduced, using

---

\*Corresponding author

3D Gaussians to represent scenes. By projecting these Gaussians onto the image plane and aggregating their contributions, 3DGS significantly accelerates rendering, achieving real-time performance while maintaining high visual fidelity.

However, both NeRF (Mildenhall et al., 2020) and 3DGS (Kerbl et al., 2023) face limitations when dealing with view-dependent effects—phenomena where the appearance of a scene changes with the viewing direction due to reflections, refractions, and complex material properties. NeRF addresses view dependency by conditioning the radiance on viewing direction, but capturing high-frequency specular highlights and anisotropic reflections remains a challenge. Similarly, 3DGS, constrained by its purely spatial (3D) representation, struggles to model these effects accurately, especially in scenes with glossy surfaces, transparency, or significant anisotropy.

To overcome these challenges, recent work by Diolatzis et al. (2024) introduced N-dimensional Gaussians (N-DG), extending the Gaussian representation into higher dimensions by incorporating additional variables such as viewing direction, leading to a 6D spatial-angular representation. This approach allows for a more expressive model that can capture view-dependent effects, enabling more accurate rendering of complex visual phenomena like specular reflections and refractions. The 6D representation combines position and direction, effectively modeling how light interacts with surfaces from different viewpoints.

Despite these advancements, N-DG presents certain drawbacks. The 6D Gaussian representation and its optimization scheme are suboptimal, resulting in rendering inefficiencies. Specifically, the method tends to allocate a significantly higher number of Gaussian points in scenes with strong view dependency to achieve acceptable rendering quality. Conversely, in scenes without substantial view-dependent effects, it may under-utilize resources, leading to fewer Gaussian points and potential loss of detail. This imbalance affects both the efficiency and the scalability of the method, making it less practical for a wide range of applications.

In this paper, we propose *6D Gaussian Splatting (6DGS)*, a novel method that integrates the strengths of both 3DGS and N-DG while addressing their respective limitations. First, we enhance the handling of color and opacity within the Gaussians. By refining the representation of these properties, our method more effectively captures the view-dependent effects. This leads to a more accurate depiction of scenes with transparent or glossy materials and intricate lighting conditions. Additionally, we develop an optimization scheme that leverages the additional directional information available in the 6D representation, which allows for better adaptive control of Gaussian distribution across the scene.

Our 6DGS method is designed to be fully compatible with the existing 3DGS framework. This compatibility ensures that applications and systems currently using 3DGS can adopt our method with minimal modifications, allowing for seamless integration and immediate benefits in rendering quality and performance. Furthermore, we provide a comprehensive theoretical analysis of the conditional Gaussian parameters derived from the 6D representation, elucidating their physical significance in the context of rendering and offering insights into how our method models the interaction of light with surfaces from different viewing directions.

We validate our method through extensive experiments on two datasets: a custom dataset with physically based rendering using ray tracing, named the *6DGS-PBR* dataset, and a public dataset without strong view-dependent effects, *i.e.*, the Synthetic NeRF dataset (Mildenhall et al., 2020). The 6DGS-PBR dataset contains scenes with complex geometries, materials, and lighting conditions that exhibit strong view-dependent effects, making it suitable for testing our method's capabilities in handling complex light interactions. Our results demonstrate that 6DGS significantly outperforms existing methods in both rendering quality and efficiency on the 6DGS-PBR dataset and achieves comparable performance on the Synthetic NeRF dataset. Specifically, we achieve up to a 15.73 dB improvement in peak signal-to-noise ratio (PSNR) while using only 33.5% of the Gaussian points compared to 3DGS. These improvements underscore the effectiveness of our approach in capturing fine details and complex view-dependent effects in real-time rendering scenarios.

Our contributions can be summarized as follows:

- **Enhanced 6D Gaussian Representation and Optimization**: We propose *6D Gaussian Splatting (6DGS)*, which advances the 6D Gaussian representation by improving color and opacity modeling and introducing an optimization strategy that leverages directional information. These enhancements result in superior rendering quality with fewer Gaussian points, particularly in scenes exhibiting complex view-dependent effects.

- **Validation**: We validate our 6DGS method on a custom physically-based ray tracing dataset (*6DGS-PBR*), demonstrating its superiority in both image quality and rendering speed compared to existing approaches. Additionally, our evaluation on a public dataset showcases its generalizability to scenes without strong view-dependent effects.
- **Compatibility with the 3DGS Framework**: Our 6DGS method is compatible with the existing 3DGS optimization framework, allowing applications to adopt our method with minimal modifications and benefit from enhanced performance without extensive changes.
- **Theoretical Analysis**: We provide a theoretical analysis of the conditional Gaussian parameters derived from the 6D representation, highlighting their physical significance in rendering and how they contribute to modeling view-dependent effects.

## 2 RELATED WORK

The field of novel view synthesis has seen significant advancements in recent years, with various techniques developed to enhance rendering quality and efficiency. This section reviews the most relevant works on 3D Gaussian splatting, N-dimensional Gaussians, and Gaussian-based ray tracing.

**3D Gaussian Splatting.** 3D Gaussian splatting (3DGS) (Kerbl et al., 2023) has emerged as a significant advancement in computer graphics and 3D vision, achieving high-fidelity rendering quality while maintaining real-time performance. Numerous works have been proposed to improve rendering quality (Yu et al., 2024; Lu et al., 2024), rendering efficiency (Lee et al., 2024; Bagdasarian et al., 2024), and training optimization (Kheradmand et al., 2024; Höllein et al., 2024), as well as to explore applications (Kocabas et al., 2024; Zhou et al., 2024b; Niedermayr et al., 2024) and extensions (Charatan et al., 2024; Luiten et al., 2024; Wu et al., 2024; Tang et al., 2024; Gao et al., 2024) of 3DGS. For example, Mip-Splatting (Yu et al., 2024) introduces a Gaussian low-pass filter based on Nyquist's theorem to address aliasing and dilation artifacts by matching the maximal sampling rate across all observed samples. Compact3D (Lee et al., 2024) applies vector quantization to compress different attributes into corresponding codebooks, storing the index of each Gaussian to reduce storage overhead. 3DGS-MCMC (Kheradmand et al., 2024) proposes densification and pruning strategies in 3DGS as deterministic state transitions of Markov Chain Monte Carlo (MCMC) samples instead of using heuristics. Many of these advancements and applications of 3DGS can potentially be applied to our 6DGS, which extends 3DGS by incorporating an additional directional component.

**N-dimensional Gaussians.** To enhance the 3D Gaussian representation, researchers have introduced other Gaussian representations for rendering. 2DGS (Huang et al., 2024) introduces a perspective-accurate 2D splatting process utilizing ray-splat intersection and rasterization to enhance geometry reconstruction. 3D-HGS (Li et al., 2024) proposes a 3D half-Gaussian kernel to improve performance without compromising rendering speed. 4DGS (Yang et al., 2024) proposes to approximate the underlying spatio-temporal 4D volume of a dynamic scene by optimizing a collection of 4D primitives. N-DG (Diolatzis et al., 2024) introduces N-dimensional Gaussians (N-DG) along with a high-dimensional culling scheme inspired by locality-sensitive hashing. Specifically, N-DG introduces a 10-dimensional Gaussian (10-DG) that incorporates geometry and material information such as world position, view direction, albedo, and roughness, as well as a 6-dimensional Gaussian (6-DG) that includes world position and view direction. Our 6DGS combines the strengths of 3DGS and N-DG for better representation and better adaptive control of Gaussians.

**Gaussian-based Ray Tracing.** Unlike most 3DGS methods that render Gaussians via rasterization, some approaches have proposed Gaussian-based ray tracing, generally at the cost of slower rendering speeds or even lower quality. For instance, Condor et al. (2024) models scattering and emissive media using mixtures of simple kernel-based volumetric primitives but achieve lower quality and slower speeds compared to 3DGS. Blanc et al. (2024) enables differentiable ray casting of irregularly distributed Gaussians using a BVH structure, but their rendering is slower than rasterization-based methods. Moenne-Loccoz et al. (2024) performs ray tracing with BVH for secondary lighting effects such as shadows and reflections, but this approach is approximately three times slower than rasterization. Zhou et al. (2024a) proposes a unified rendering primitive based on 3D Gaussian distributions, enabling physically based scattering for accurate global illumination but do not achieve real-time performance. In contrast, our 6DGS method slices 6D Gaussians into conditional 3D Gaussians and renders via rasterization, approximating physically based ray-traced images with high fidelity while achieving real-time performance.

## 3 PRELIMINARY

### 3.1 6D GAUSSIAN REPRESENTATION

N-dimensional Gaussian (N-DG) (Diolatzis et al., 2024) extends the 3DGS (Kerbl et al., 2023) approach by introducing a 6D Gaussian representation, which includes both position and directional information. Specifically, each Gaussian is defined by the following parameters: position ($\mu_p \in \mathbb{R}^3$), direction ($\mu_d \in \mathbb{R}^3$), covariance matrix ($\Sigma \in \mathbb{R}^{6 \times 6}$), opacity ($\alpha \in \mathbb{R}^1$), and color ($c \in \mathbb{R}^3$).

The covariance matrix $\Sigma$ is modeled as a 6D matrix that includes both spatial and directional variances. To ensure stability and positive definiteness, we utilize a Cholesky decomposition, parameterizing $\Sigma$ with a lower triangular matrix $L$ as $\Sigma = LL^\top$. Diagonal elements are ensured to be positive using an exponential activation function, while the off-diagonal elements are constrained within $[-1, 1]$ using a sigmoid function.

### 3.2 SLICE 6D GAUSSIAN TO CONDITIONAL 3DGS

The slicing Gaussians technique is adopted to render 6D Gaussians efficiently. For a given viewing direction $d$, we compute a conditional 3D Gaussian that represents the slice of our 6D Gaussian in that direction. Let $X = [X_p, X_d]$ be the 6D Gaussian random variable, where $X_p$ represents the position and $X_d$ represents the direction. The joint distribution is:

$$X \sim \mathcal{N}\left(\begin{bmatrix} \mu_p \\ \mu_d \end{bmatrix}, \begin{bmatrix} \Sigma_p & \Sigma_{pd} \\ \Sigma_{pd}^\top & \Sigma_d \end{bmatrix}\right), \tag{1}$$

where $\Sigma_{pd}$ represents the cross-covariance between position and direction.

For rendering, we compute the conditional 3D Gaussian distribution $p(X_p|X_d = d)$, using the properties of multivariate Gaussians, as follows:

$$p(X_p|X_d = d) = \mathcal{N}(\mu_{\text{cond}}, \Sigma_{\text{cond}}), \tag{2}$$

where

$$\mu_{\text{cond}} = \mu_p + \Sigma_{pd}\Sigma_d^{-1}(d - \mu_d), \tag{3}$$
$$\Sigma_{\text{cond}} = \Sigma_p - \Sigma_{pd}\Sigma_d^{-1}\Sigma_{pd}^\top. \tag{4}$$

The opacity of each Gaussian also depends on the view direction as:

$$f_{\text{cond}} = \exp\left(-(d - \mu_d)^\top \Sigma_d^{-1}(d - \mu_d)\right), \tag{5}$$
$$\alpha_{\text{cond}} = \alpha \cdot f_{\text{cond}}, \tag{6}$$

where $f_{\text{cond}}$ represents the conditional probability density function (PDF) of the directional component, evaluated at the current viewing direction $d$, and $\alpha_{\text{cond}}$ is the modulated opacity based on $f_{\text{cond}}$.

## 4 METHOD

### 4.1 THEORETICAL ANALYSIS OF CONDITIONAL GAUSSIAN

This section provides a theoretical analysis of the conditional Gaussian parameters derived from the 6D Gaussian representation, highlighting their physical meanings in Gaussian splatting.

**Conditional Mean ($\mu_{\text{cond}}$).** The conditional mean $\mu_{\text{cond}}$ can be interpreted as the Best Linear Unbiased Estimator (BLUE) for the position component $X_p$, as in Equation 3. A linear estimator $\hat{\delta}$ is BLUE if it is unbiased ($E[\hat{\delta}] = \delta$) and has the smallest variance among all linear unbiased estimators. The expression for $\mu_{\text{cond}}$ given earlier ensures that the position mean is unbiased and minimizes variance according to the Schur complement. The conditional mean $\mu_{\text{cond}}$ represents the expected position of the Gaussian splat in 3D space, adjusting dynamically based on the viewing direction to capture non-planar geometry and parallax effects.

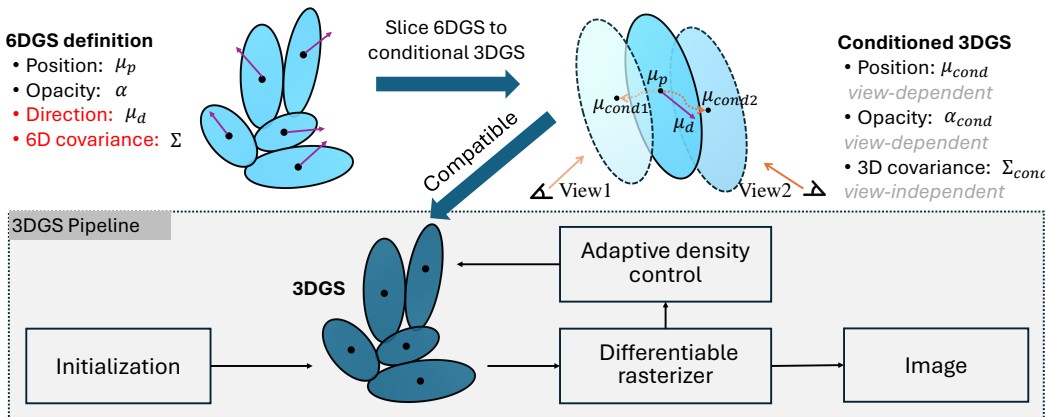

Figure 2: Proposed method of direction-aware 6DGS compatible with the existing 3DGS pipeline. The position and opacity of the conditional 3D Gaussian are adjusted according to the view direction.

**Conditional Covariance ($\Sigma_{\text{cond}}$).** The conditional covariance $\Sigma_{\text{cond}}$ is derived as the Schur complement of $\Sigma_d$ in the joint covariance matrix $\Sigma$, as in Equation 4. This covariance matrix represents the residual uncertainty in $X_p$ after accounting for the correlation with $X_d$. Notably, $\Sigma_{\text{cond}}$ remains constant regardless of the viewing direction $d$. The conditional covariance $\Sigma_{\text{cond}}$ describes the shape and orientation of the Gaussian, encoding the local surface geometry and the uncertainty in $X_p$ after accounting for the correlation with $X_d$.

**Conditional Opacity ($\alpha_{\text{cond}}$).** The function $f_{\text{cond}}$ is derived from the conditional probability density function (PDF) of the directional component in the 6D Gaussian. Specifically, $f_{\text{cond}}$ represents the likelihood of the viewing direction $d$ aligning with the mean direction $\mu_d$ given the directional variance $\Sigma_d$. Equation 5 is a standard form of the exponent in a Gaussian PDF, reflecting the Mahalanobis distance. The opacity $\alpha_{\text{cond}}$ is scaled by $f_{\text{cond}}$ in Equation 6 to introduce view-dependency, based on the principle that visibility may vary with viewing direction due to anisotropic properties. By modulating the base opacity $\alpha$ with $f_{\text{cond}}$, we ensure that $\alpha_{\text{cond}}$ dynamically adjusts to reflect the directional characteristics of the Gaussian splat. This approach leverages the mathematical relationship between conditional probabilities and Gaussian distributions, enabling realistic and physically plausible rendering of view-dependent effects.

Together, these parameters form a robust framework that integrates view-dependent and view-independent characteristics, enabling efficient and realistic rendering, particularly effective for scenes with varied geometries and materials.

### 4.2 Enhanced 6D Gaussian Representation

In N-DG (Diolatzis et al., 2024), the color $c$ is defined by learnable RGB values. To introduce view-dependent effects, we adopt the spherical harmonics representation $\phi_\beta(d) : \mathbb{R}^3 \to \mathbb{R}^3$, as used in 3DGS. This representation captures the variation in color based on the viewing direction. The spherical harmonics functions $Y_\ell^m(d)$ of order $\ell = 3$ are parameterized by the coefficients $\beta \in \mathbb{R}^{48}$. The view-dependent spherical harmonics representation can be expressed as:

$$\phi_\beta(d) = f\left(\sum_{\ell=0}^{\ell_{\max}} \sum_{m=-\ell}^{\ell} \beta_\ell^m Y_\ell^m(d)\right), \tag{7}$$

where $f$ is the sigmoid function used to normalize the colors.

To better control the effect of the view direction on opacity, we refine the conditional PDF of the directional component as follows:

$$f_{\text{cond}} = \exp\left(-\lambda_{\text{opa}} \cdot (d - \mu_d)^\top \Sigma_d^{-1} (d - \mu_d)\right), \tag{8}$$

---

**Algorithm 1** Slice 6DGS to 3DGS. In inference, we pre-compute $\Sigma_{\text{cond}}$, and the scale $S$ and rotation $R$ are not required. Only $\mu_{\text{cond}}$ and $\alpha_{\text{cond}}$ (highlighted in blue) need to be computed for each rendering.

---

**Input:** Lower triangular $L$, position $\mu_p$, direction $\mu_d$, opacity $\alpha$, color $c$, view direction $d$
**Output:** Conditional position, $\mu_{\text{cond}}$, covariance, $\Sigma_{\text{cond}}$, opacity $\alpha_{\text{cond}}$, scale $S$, rotation $R$

1: **Compute covariance matrix as**: $\Sigma = LL^\top$

2: **Partition $\Sigma$ into blocks**: $\Sigma = \begin{bmatrix} \Sigma_p & \Sigma_{pd} \\ \Sigma_{pd}^\top & \Sigma_d \end{bmatrix}$

3: **Calculate the conditional covariance, mean, and opacity**:

$$\Sigma_{\text{cond}} = \Sigma_p - \Sigma_{pd}\Sigma_d^{-1}\Sigma_{pd}^\top$$

$$\mu_{\text{cond}} = \mu_p + \Sigma_{pd}\Sigma_d^{-1}(d - \mu_d)$$

$$\alpha_{\text{cond}} = \alpha \cdot f_{\text{cond}}, \text{ where } f_{\text{cond}} = \exp\left(-\lambda_{\text{opa}} \cdot (d - \mu_d)^\top \Sigma_d^{-1}(d - \mu_d)\right)$$

4: **Perform SVD as**: $\Sigma_{\text{cond}} = UDU^\top$
5: **Extract rotation matrix and scale**: $R = U, \quad S = \sqrt{\text{diag}(D)}, \quad R_{:,3} = R_{:,3} \cdot \text{sign}(\det(R))$

---

where $0 < \lambda_{\text{opa}} < 1$ is a hyper-parameter or per-Gaussian learnable parameter that controls the influence of the view direction on the opacity. When adjusting $\lambda_{\text{opa}}$ as a hyper-parameter, we can modulate the resulting conditional opacity $\alpha_{\text{cond}}$, thereby controlling the density of the Gaussian during optimization; when treating $\lambda_{\text{opa}}$ as a per-Gaussian learnable parameter, it adapts the level of view-dependency for each Gaussian through training. This allows for finer control over the rendering process, enabling the representation of more complex visual effects.

### 4.3 Improved Control of Gaussians

To enhance the control of Gaussians, we adapt the explicit adaptive control mechanism from 3DGS, leveraging the additional directional information available in our 6D Gaussian representation. Instead of relying on the high-dimensional culling scheme used in N-DG, our approach focuses on refining Gaussian placement and density based on the scene's geometry.

In 3DGS, the adaptive Gaussian densification scheme involves two primary operations: cloning and splitting. Cloning is employed when small-scale geometry is insufficiently covered by existing Gaussians, while splitting is used to divide a large Gaussian into smaller ones when it encompasses fine details of the geometry. This scheme requires the scale and rotation of each Gaussian, which are not directly provided in our 6D Gaussian representation.

To extract the necessary scale and rotation information from our 6D representation, we utilize the conditional covariance matrix $\Sigma_{\text{cond}}$. By performing Singular Value Decomposition (SVD) on $\Sigma_{\text{cond}}$, we can decompose the matrix into its principal components, revealing both the rotation and scale of the Gaussian. The decomposition is given by $\Sigma_{\text{cond}} = UDU^\top$, where $U$ is an orthogonal matrix, and $D$ is a diagonal matrix containing the singular values. From this decomposition: the rotation matrix $R$ is derived from the left singular vectors as $R = U$, and the scale vector $S$ is obtained by taking the square root of the diagonal elements of $D$ as $S = \sqrt{\text{diag}(D)}$. To ensure that the rotation matrix $R$ forms a right-handed coordinate system, we adjust its last column based on the sign of its determinant: $R_{:,3} = R_{:,3} \cdot \text{sign}(\det(R))$.

This SVD-based decomposition allows us to represent the conditional Gaussian as an oriented ellipsoid, with clearly defined rotation and scale. By extracting these components, we can directly apply the adaptive Gaussian densification scheme from 3DGS (Kerbl et al., 2023), thereby improving the coverage of small-scale geometry and enhancing the overall quality of the rendered scene.

In 3DGS, Gaussians are pruned when their opacity falls below a minimum threshold $\tau_{\text{min}}$ (*e.g.*, $\tau_{\text{min}} = 0.005$ by default) or when they become excessively large. In our 6DGS approach, the conditional opacity $\alpha_{\text{cond}}$ is also influenced by the opacity parameter $\lambda_{\text{opa}}$. By incorporating $\lambda_{\text{opa}}$, we can fine-tune the density of Gaussians with greater precision, allowing for more granular control over which Gaussians are retained or pruned based on their opacity.

### 4.4 COMPATIBILITY WITH 3DGS

Algorithm 1 outlines the implementation details for converting (*i.e.*, slicing) our 6DGS representation into a 3DGS-compatible format in a single function. Once this slicing operation is performed, the subsequent implementation remains identical to that of 3DGS.

The proposed 6DGS seamlessly integrates with the existing training framework of 3DGS, including the use of the same loss functions, optimizers, and training hyperparameters (except for the minimum opacity threshold where we set $\tau_{\min} = 0.01$). By slicing 6DGS into a conditional 3DGS format, we can directly utilize the adaptive density control and the differentiable rasterization method employed in 3DGS. This compatibility ensures that many downstream applications can effortlessly switch from 3DGS to our 6DGS, resulting in enhanced performance without the need for extensive modifications.

Note that the scale $S$ and rotation $R$ are only required during refinement iterations (*e.g.*, every 100 iterations) for the adaptive density control. Additionally, during inference, $\Sigma_{\text{cond}}$ can be pre-computed. Therefore, only $\mu_{\text{cond}}$ and $\alpha_{\text{cond}}$ need to be computed for each rendering. To further enhance rendering efficiency, the slice operation can be implemented in CUDA.

## 5 EXPERIMENTS

### 5.1 EXPERIMENTAL PROTOCOL

**Datasets.** We evaluate 6DGS on two datasets: the public Synthetic NeRF dataset (Mildenhall et al., 2020) and a custom dataset using physically-based rendering (PBR), which we refer to as the 6DGS-PBR dataset. The 6DGS-PBR dataset consists of six scenes: 1) `cloud` from the Walt Disney Animation Studios volumetric cloud dataset [1]; 2) `bunny-cloud`, `explosion`, and `smoke` from OpenVDB volumetric models [2]; 3) `suzanne`, the standard Blender test mesh, with a "*Glass BSDF*" translucent material applied to it; 4) `ct-scan`, prepared from a real CT scan.

We rendered these scenes in Blender using its PBR engine "Cycle". At the image sizes (with width equal to height) as listed in Table 2 on an NVIDIA Tesla V100 GPU, the rendering times per view were as follows: `bunny-cloud` took 504.0 seconds per view, `cloud` 838.6 seconds, `explosion` 35.5 seconds, `smoke` 71.5 seconds, `suzanne` 9.1 seconds, and `ct-scan` 28.5 seconds. For the `ct-scan` object, we generated 360 views with corresponding camera poses and randomly selected 324 images for training and 36 for testing. For each of the other objects, we rendered 150 views, randomly selecting 100 images for training and 50 for testing. We will make our 6DGS-PBR dataset publicly available to the community.

**Evaluation Metrics.** We evaluate our method's performance using Peak Signal-to-Noise Ratio (PSNR) and Structural Similarity Index Measure (SSIM) for image quality, number of Gaussian points (# point) for size, and Frames Per Second (FPS) for rendering speed.

**Implementation.** In our experiments, we set $\lambda_{\text{opa}} = 0.35$ and the minimum opacity threshold $\tau = 0.01$. For learnable $\lambda_{\text{opa}}$, we initialize $\lambda_{\text{opa}} = 0.35$ and make it trainable only during the iterations of 15,000 - 28,000. All other parameters are set to their default values as in 3DGS (Kerbl et al., 2023). For the `ct-scan` object, we initialize the point cloud using the marching cubes algorithm as in DDGS (Gao et al., 2024). For the other objects and the Synthetic NeRF dataset (Mildenhall et al., 2020), we randomly initialize the point cloud with 100,000 points within a cube encompassing the scene. Training is performed on a single NVIDIA Tesla V100 GPU with 16 GB of memory, using the Adam optimizer (Kingma & Ba, 2014). We set the learning rate to $1 \times 10^{-2}$ for the 6D covariance parameters and $1 \times 10^{-3}$ for the direction component $\mu_d$. The default learning rates from 3DGS are applied to the remaining parameters.

### 5.2 COMPARISON WITH STATE-OF-THE-ART

Table 1 compares our 6DGS method with 3DGS and N-DG on the 6DGS-PBR dataset. Our 6DGS achieves significantly better image quality, with an average improvement of **+10.08 dB** in PSNR,

---

[1] https://disneyanimation.com/resources/clouds/
[2] https://www.openvdb.org/download/

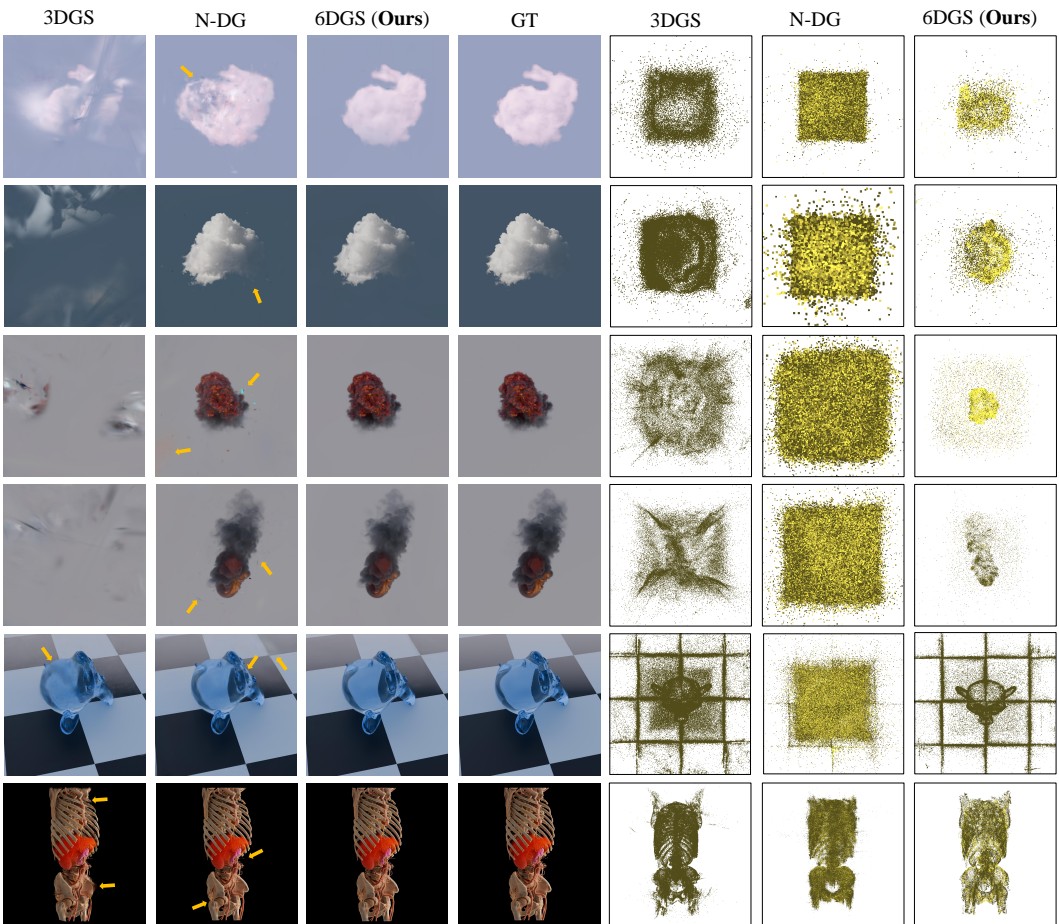

Figure 3: Qualitative comparison of methods on the 6DGS-PBR dataset (zoom in for details).

Table 1: Comparison of methods on the 6DGS-PBR dataset.

| | 3DGS | | | N-DG | | | 6DGS (Ours) | | |
|---|---|---|---|---|---|---|---|---|---|
| | PSNR↑ | SSIM↑ | # points↓ | PSNR↑ | SSIM↑ | # points↓ | PSNR↑ | SSIM↑ | # points↓ |
| bunny-cloud | 30.75 | 0.988 | 21,074 | 35.48 | 0.990 | 530,711 | **41.57** | **0.993** | **6,660** |
| cloud | 29.70 | 0.972 | 58,233 | **42.40** | **0.991** | 98,149 | 40.41 | 0.991 | **12,657** |
| explosion | 26.75 | 0.953 | 51,140 | 40.16 | 0.989 | 207,778 | **42.48** | **0.991** | **17,133** |
| smoke | 28.55 | 0.969 | 60,533 | **41.61** | **0.992** | 212,050 | 40.61 | **0.992** | **10,762** |
| suzanne | 23.70 | 0.901 | 270,001 | 26.00 | 0.921 | 232,145 | **27.03** | **0.928** | **174,746** |
| ct-scan | 25.71 | 0.917 | 229,683 | 30.96 | 0.952 | 1,073,082 | **33.56** | **0.965** | **182,981** |
| avg | 27.53 | 0.950 | 115,111 | 36.10 | 0.973 | 392,319 | **37.61** | **0.977** | **67,490** |

while using significantly fewer Gaussian points (a reduction of **41.4%** on average) compared to 3DGS. Compared to N-DG, our 6DGS also achieves higher image quality (**+1.51 dB** on average) and uses significantly fewer Gaussian points (a reduction of **82.8%** on average).

Table 2 presents the rendering speed comparison. In the FPS comparison, we calculated the average FPS from 20 test images, where each image's FPS value is the mean of 500 repeated measurements. Because our 6DGS uses fewer Gaussian points, it achieves faster rendering speeds compared to both 3DGS and N-DG. We further integrate FlashGS (Feng et al., 2024), an open-source CUDA Python library for efficient differentiable rasterization of 3D Gaussian splatting, into our 6DGS rendering pipeline, referred to as 6DGS-flash. With 6DGS-flash, we achieve an average rendering speed of **326.3 FPS**, which is sufficient for many real-time applications. Note that, we can further improve our rendering speed by implementing our 6DGS slicing algorithm in CUDA.

Table 2: Rendering speed (FPS) comparison of methods on the 6DGS-PBR dataset.

|  | bunny-cloud | cloud | explosion | smoke | suzanne | ct-scan | avg |
|---|---|---|---|---|---|---|---|
| Image size | 1408 | 1408 | 1024 | 1536 | 1408 | 1024 | N/A |
| 3DGS | 49.1 | 74.4 | **161.9** | 96.7 | 42.0 | 259.7 | 114.0 |
| N-DG | 27.0 | 90.7 | 65.6 | 50.4 | **46.9** | 13.3 | 49.0 |
| **6DGS (Ours)** | **178.4** | **178.2** | 120.2 | **138.6** | 34.7 | **276.8** | **154.5** |
| **6DGS-flash (Ours)** | 315.3 | 318.0 | 367.7 | 345.5 | 295.9 | 315.1 | 326.3 |

Table 3: Comparison of methods on the Synthetic NeRF dataset.

|  | 3DGS | | | N-DG | | | **6DGS (ours)** | | |
|---|---|---|---|---|---|---|---|---|---|
|  | PSNR↑ | SSIM↑ | # points↓ | PSNR↑ | SSIM↑ | # points↓ | PSNR↑ | SSIM↑ | # points↓ |
| chair | **35.91** | **0.987** | 272,130 | 30.87 | 0.956 | **108,091** | 35.49 | 0.986 | 223,747 |
| drums | 26.15 | **0.955** | 346,245 | 24.37 | 0.927 | **106,756** | **26.45** | 0.953 | 250,267 |
| ficus | **34.49** | **0.987** | 295,997 | 29.82 | 0.965 | **59,052** | 33.45 | 0.984 | 197,741 |
| hotdog | 37.72 | **0.985** | 147,098 | 33.89 | 0.971 | **82,261** | **37.90** | **0.985** | 102,451 |
| lego | **35.79** | **0.983** | 322,704 | 29.85 | 0.948 | 151,291 | 35.25 | 0.980 | 233,227 |
| materials | 29.98 | 0.960 | 282,334 | 26.86 | 0.938 | **77,206** | **30.71** | **0.967** | 222,209 |
| mic | 35.47 | **0.992** | 310,608 | 29.99 | 0.968 | **40,848** | **36.13** | **0.992** | 272,052 |
| ship | 30.52 | **0.905** | 328,053 | 26.35 | 0.862 | 337,294 | **30.72** | 0.903 | **270,163** |
| avg | 33.25 | **0.969** | 288,146 | 29.00 | 0.942 | **120,350** | **33.26** | **0.969** | 221,482 |

In Figure 3, we compare our 6DGS method with 3DGS and N-DG in terms of rendering image quality and visualization of the generated point clouds. We observe that 3DGS fails on the `bunny-cloud`, `cloud`, `explosion`, and `smoke` datasets, exhibiting strong visible artifacts. N-DG tends to produce artifacts in the background and generates blurrier images away from the center. In contrast, our 6DGS achieves the best visual quality, closely matching the ground-truth images.

Furthermore, the visualization of the point clouds reveals that both 3DGS and N-DG struggle to generate faithful shapes of the objects. Specifically, 3DGS fails to capture the shapes in the `bunny-cloud`, `cloud`, `explosion`, and `smoke` datasets. N-DG tends to over-generate point clouds from the initialization, resulting in shapes resembling the initial marching-cube shape for `ct-scan` and a randomly sampled cube for other objects. In contrast, our 6DGS successfully reconstructs the shapes of the objects.

To evaluate the generalizability of our 6DGS method to scenes without significant view-dependent effects, we conducted experiments on the public Synthetic NeRF dataset (Mildenhall et al., 2020), as shown in Table 3. Our results demonstrate that 6DGS achieves image quality comparable to 3DGS while using significantly fewer Gaussian points (a reduction of **23.1%** on average). In contrast, N-DG produces considerably lower image quality and uses fewer Gaussian points because it is specifically designed to model scenes with view-dependent effects and tends to underperform in scenes without strong view dependency.

## 5.3 ABLATION STUDY

We conduct ablation experiments on both the 6DGS-PBR dataset and Synthetic NeRF dataset (see Appendix Table 8) to investigate the effects of various components in our 6DGS method. Specifically, we examine the impact of the parameter $\lambda_{\text{opa}}$ in the conditional probability density function (PDF) of the directional component (see Appendix Figure 5), defined as $f_{\text{cond}} = \exp(-\lambda_{\text{opa}} \cdot D)$, where $D = (d - \mu_d)^\top \Sigma_d^{-1} (d - \mu_d)$ represents the Mahalanobis distance between the viewing direction $d$ and the Gaussian mean direction $\mu_d$, and $\lambda_{\text{opa}}$ is a scalar between 0 and 1 that controls the influence of the view direction on opacity. Our ablation experiments include several settings related to $\lambda_{\text{opa}}$:

- **No-$f_{\text{cond}}$**: Setting $\lambda_{\text{opa}} = 0$ results in $f_{\text{cond}} = \exp(0) = 1$, meaning opacity is not modulated by the view direction.
- **No-$\lambda_{\text{opa}}$**: Setting $\lambda_{\text{opa}} = 1$ uses the default $f_{\text{cond}}$ as in N-DG, $i.e.$, $f_{\text{cond}} = \exp(-D)$.
- **$\lambda_{\text{opa}} = 0.35$**: We set $\lambda_{\text{opa}} = 0.35$, so $f_{\text{cond}} = \exp(-0.35 \cdot D)$, providing a balance between no modulation and full modulation.

Table 4: Ablation study on the 6DGS-PBR dataset.

| | | bunny-cloud | cloud | explosion | smoke | suzanne | ct-scan | avg |
|---|---|---|---|---|---|---|---|---|
| **PSNR** | No-SH | 38.13 | 37.39 | 41.05 | 38.52 | 26.76 | 33.27 | 35.85 |
| | No-$f_{opa}$ | 38.15 | 36.16 | 35.49 | 35.52 | 24.88 | 29.65 | 33.31 |
| | No-$\lambda_{opa}$ | 39.12 | 39.86 | 40.55 | 36.88 | 26.92 | 32.99 | 36.05 |
| | $\tau = 0.005$ | 40.15 | 40.46 | **43.14** | **41.04** | 27.09 | 33.45 | 37.56 |
| | $\lambda_{opa} = 0.35$ | 40.47 | **40.73** | 42.69 | 40.45 | **27.15** | 33.42 | 37.49 |
| | learnable-$\lambda_{opa}$ | **41.57** | 40.42 | 42.48 | 40.61 | 27.03 | **33.56** | **37.61** |
| **# points** | No-SH | **6,196** | **11,356** | 16,465 | **9,929** | **171,472** | 177,204 | 65,437 |
| | No-$f_{cond}$ | 11,698 | 31,999 | 39,736 | 35,023 | 349,716 | 320,879 | 131,509 |
| | No-$\lambda_{opa}$ | 4,860 | 11,736 | **13,582** | 10,540 | 158,041 | **154,997** | **58,959** |
| | $\tau = 0.005$ | 63,042 | 51,883 | 61,507 | 49,422 | 301,614 | 405,332 | 155,467 |
| | $\lambda_{opa} = 0.35$ | 6,830 | 12,454 | 17,051 | 10,570 | 172,373 | 181,539 | 66,803 |
| | learnable-$\lambda_{opa}$ | 6,660 | 12,657 | 17,133 | 10,762 | 174,746 | 182,981 | 67,490 |

- **Learnable $\lambda_{opa}$**: $\lambda_{opa}$ is treated as a trainable parameter between 0 and 1 for each Gaussian, allowing the model to learn the optimal degree of view-dependent opacity.

Additionally, we include ablations of other components:

- **No-SH**: Color is treated as a learnable RGB parameter per Gaussian, as in N-DG, instead of using spherical harmonics (SH) to represent view-dependent color.
- **$\tau = 0.005$**: The default minimum opacity threshold used in 3DGS, whereas our default choice is $\tau = 0.01$.

These ablation studies help us understand the contribution of each component to the overall performance of our 6DGS method.

Table 4 presents the results of our ablation studies. Compared to our model with $\lambda_{opa} = 0.35$, setting *No-$f_{cond}$* (*i.e.*, $\lambda_{opa} = 0$) significantly degrades the image quality while increasing the number of Gaussian points. This highlights the importance of incorporating view dependency into the opacity function. Conversely, when we set *No-$\lambda_{opa}$* (*i.e.*, $\lambda_{opa} = 1$), the image quality also degrades, but the number of Gaussian points decreases. This suggests that adjusting $\lambda_{opa}$ allows us to balance the trade-off between image quality and the number of Gaussian points, which directly affects rendering speed. When we make $\lambda_{opa}$ a learnable parameter, the model achieves improved image quality with a comparable number of Gaussian points.

Furthermore, the *No-SH* setting, where spherical harmonics (SH) are not used for color representation, results in degraded image quality. This is because SH provides a powerful representation for view-dependent color effects. Setting the minimum opacity threshold to $\tau = 0.005$ (the default value in 3DGS) achieves similar image quality but requires significantly more Gaussian points. This is because introducing view-dependent opacity decay via the $f_{cond}$ function necessitates a larger minimum opacity threshold.

## 6 CONCLUSION

In this work, we introduce 6D Gaussian splatting (6DGS) that builds on the foundations laid by 3DGS and N-dimensional Gaussians (N-DG). By improving the handling of color and opacity within the 6D spatial-angular framework and optimizing the adaptive control of Gaussians using additional directional information, we develop a method that not only maintains compatibility with the 3DGS framework but also offers superior rendering capabilities, particularly in modeling complex view-dependent effects. Our extensive experiments on the custom physically-based rendering dataset (6DGS-PBR) demonstrate the effectiveness of 6DGS in achieving higher image quality and faster rendering speed compared to 3DGS and N-DG.

Looking ahead, 6DGS opens avenues for more accurate and efficient real-time volumetric rendering in virtual and augmented reality, gaming, and film production. Future research will explore further optimizations and extensions to enhance the scalability and robustness of the 6DGS framework, as well as its application to dynamic scenes and integration with advanced lighting models.

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

# A  APPENDIX

## A.1  REAL-WORLD SCENES

Table 5: Comparison of methods on real-world datasets: Deep Blending (Hedman et al., 2021) and Tanks & Temples (Knapitsch et al., 2017).

| Dataset | Scene | 3DGS | | | N-DG | | | 6DGS (Ours) | | |
|---|---|---|---|---|---|---|---|---|---|---|
| | | PSNR | SSIM | # points | PSNR | SSIM | # points | PSNR | SSIM | # points |
| Deep Blending | drjohnson | **29.22** | **0.898** | 3,276,989 | 26.31 | 0.828 | 494,494 | 28.12 | 0.883 | 2,074,490 |
| | playroom | **29.74** | **0.900** | 2,332,830 | 27.74 | 0.866 | 258,999 | 29.06 | 0.892 | 1,685,626 |
| | avg | **29.48** | **0.899** | 2,804,910 | 27.03 | 0.847 | 376,747 | 28.59 | 0.888 | 1,880,058 |
| Tanks& Temples | train | 21.75 | **0.803** | 1,100,525 | 13.26 | 0.440 | 996,826 | **21.95** | 0.787 | 839,408 |
| | truck | 25.08 | **0.869** | 2,606,855 | 13.65 | 0.462 | 663,616 | 25.05 | 0.859 | 2,050,162 |
| | avg | 23.42 | **0.836** | 1,853,690 | 13.46 | 0.451 | 830,221 | **23.50** | 0.823 | 1,444,785 |

Table 6: Comparison of methods on the real-world dataset: Shiny (Wizadwongsa et al., 2021)

| Scene | 3DGS | | | 6DGS (Ours) | | |
|---|---|---|---|---|---|---|
| | PSNR | SSIM | # points | PSNR | SSIM | # points |
| cd | 25.51 | 0.843 | 1,128,098 | **28.39** | **0.895** | 596,789 |
| crest | 18.80 | 0.622 | 4,611,724 | **19.35** | **0.648** | 3,144,178 |
| food | **18.39** | **0.500** | 2,362,888 | 17.97 | 0.475 | 1,190,056 |
| giants | **24.24** | **0.844** | 2,341,337 | 24.15 | 0.826 | 1,635,315 |
| lab | 24.69 | 0.836 | 843,202 | **27.66** | **0.903** | 490,878 |
| pasta | **15.45** | **0.373** | 2,287,582 | 14.90 | 0.349 | 978,188 |
| seasoning | 26.25 | **0.823** | 1,085,732 | **26.36** | 0.811 | 524,848 |
| tools | **26.20** | **0.908** | 1,180,973 | 25.28 | 0.884 | 593,190 |
| avg | 22.44 | 0.719 | 1,980,192 | **23.01** | **0.724** | 1,144,180 |

We evaluate 6DGS on three real-world datasets: Deep Blending (Hedman et al., 2021), Tanks & Temples (Knapitsch et al., 2017), and Shiny (Wizadwongsa et al., 2021) to validate its robustness and effectiveness in practical scenarios, as shown in Table 5 and 6. These datasets present diverse challenges, including scenes with varying levels of complexity, strong view-dependent effects, and detailed geometric structures.

In the Deep Blending dataset (Hedman et al., 2021), 6DGS achieves comparable quality to 3DGS, with an average PSNR of 28.59 versus 29.48, while reducing the number of Gaussian points by 33%. This demonstrates that 6DGS effectively maintains high-quality rendering even with significantly fewer points, thereby improving computational efficiency without largely compromising accuracy.

On the Tanks & Temples dataset (Knapitsch et al., 2017), which features large-scale outdoor scenes with intricate geometric details, 6DGS outperforms 3DGS with a slight improvement in PSNR (23.50 vs. 23.42) and a 22% reduction in the number of points used. These results highlight the ability of 6DGS to generalize effectively to outdoor scenes and achieve improved efficiency.

The Shiny dataset (Wizadwongsa et al., 2021), known for its challenging scenes with strong view-dependent effects, offers a rigorous test for 6DGS's directional modeling capabilities. Following the settings in NeX (Wizadwongsa et al., 2021), we resize the images to $1008 \times 567$ for cd and lab and to $1008 \times 756$ for other scenes. In this dataset, 6DGS excels, particularly in scenes such as cd and lab, where it achieves PSNR gains of +2.88 and +2.97, respectively, compared to 3DGS. Additionally, these gains are achieved while utilizing approximately 50% fewer Gaussian points. These results underscore the strength of 6DGS in handling view-dependent phenomena.

Across all datasets, the improvements in PSNR and point efficiency demonstrate the robustness and versatility of 6DGS in real-world settings. By reducing the number of Gaussian points while maintaining or exceeding rendering quality, 6DGS offers a significant advancement in computational efficiency and scalability for practical applications. This ability to model strong view-dependent effects and detailed geometry efficiently positions 6DGS as a valuable tool for various rendering tasks, from real-time applications to high-fidelity visualizations.

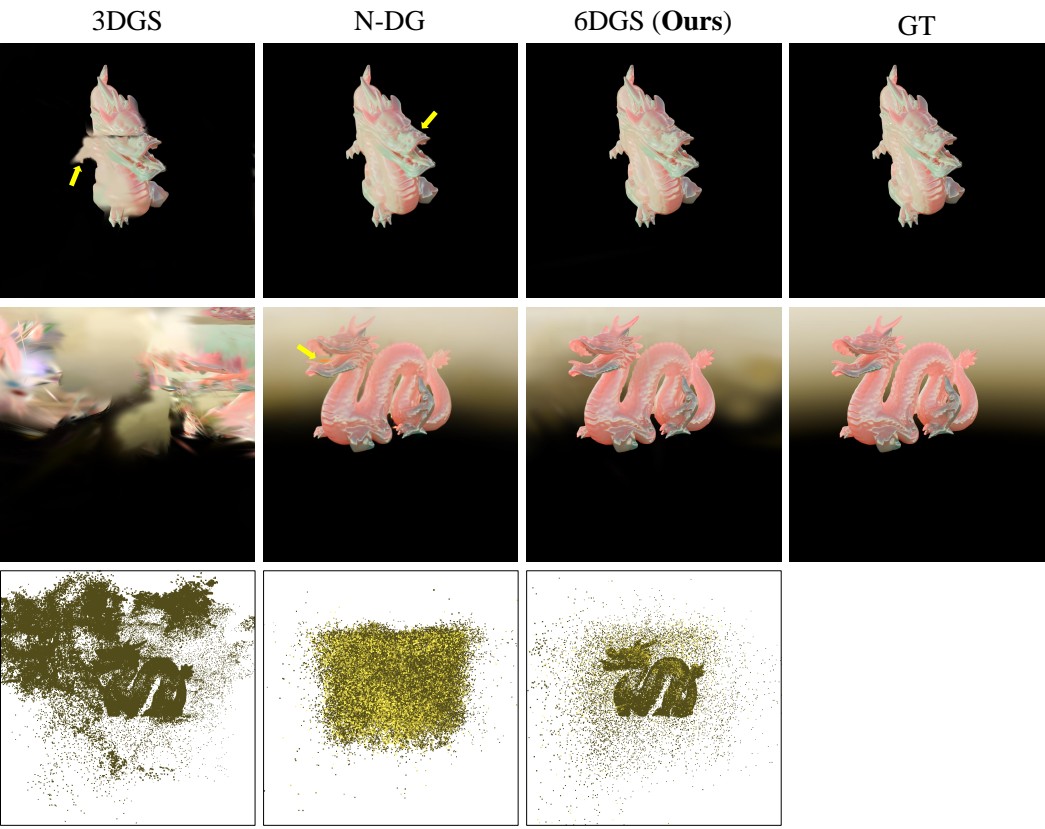

Figure 4: Qualitative comparison of methods on the subsurface scatter (SSS) dragon scene.

## A.2 SUBSURFACE SCATTERING SCENE

Table 7: Comparison of methods on the scene of subsurface scattering (SSS) dragon.

| SSS Dragon | PSNR | SSIM | # points | FPS | Train (min) |
|---|---|---|---|---|---|
| 3DGS | 26.57 | 0.813 | 269,250 | 104.6 | **24** |
| N-DG | 33.19 | 0.936 | 196,645 | 86.0 | 50 |
| **6DGS (Ours)** | **35.00** | **0.937** | **128,748** | **111.5** | 33 |
| **6DGS-Flash (Ours)** | - | - | - | **324.3** | - |

To further validate the versatility of our method, we have added a new dragon scene that exhibits both subsurface scattering (SSS) and surface scattering effects. Following the same evaluation protocol as other scenes (except for `ct-scan`), we rendered 150 views and randomly selected 100 images for training and 50 for testing. As shown in Table 7, 6DGS significantly outperforms both baselines, achieving the best PSNR (35.00) and SSIM (0.937) while using substantially fewer Gaussian points. Our method also maintains competitive training efficiency (33 minutes vs. 24 minutes for 3DGS and 50 minutes for N-DG) while achieving real-time rendering performance (111.5 FPS, further accelerated to 324.3 FPS with 6DGS-Flash). Figure 4 demonstrates that 6DGS achieves superior visual quality and more faithful geometry reconstruction, further validating our method's effectiveness in handling diverse scenarios involving both volumetric and surface-based light transport phenomena.

## A.3 VIEW-DEPENDENT OPACITY

Figure 5 illustrates the effect of adjusting the parameter $\lambda_{\text{opa}}$ on the conditional probability density function (PDF) $f_{\text{cond}}$ of the directional component, plotted as a function of the Mahalanobis distance $D$ between the viewing direction $d$ and the Gaussian mean direction $\mu_d$. The Mahalanobis distance is

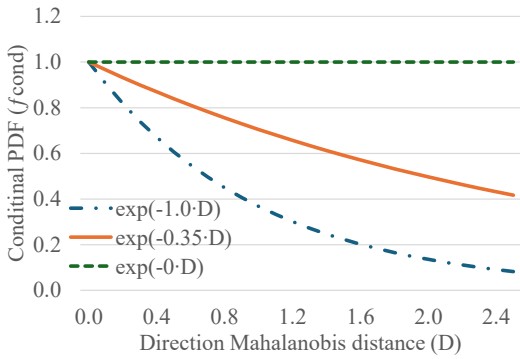

Figure 5: Effect of adjusting $\lambda_{\mathrm{opa}}$ on the conditional probability density function (PDF) of the directional component $f_{\mathrm{cond}}$, plotted as the functions of the Mahalanobis distance $D$ between the view direction $d$ and the Gaussian mean direction $\mu_d$.

defined as:

$$D = (d - \mu_d)^\top \Sigma_d^{-1} (d - \mu_d), \tag{9}$$

and the conditional PDF is given by:

$$f_{\mathrm{cond}} = \exp\left(-\lambda_{\mathrm{opa}} \cdot D\right). \tag{10}$$

When $\lambda_{\mathrm{opa}} = 1$, the Mahalanobis distance $D$ has the maximum influence on the opacity, meaning that even small deviations in the viewing direction $d$ from the Gaussian mean direction $\mu_d$ significantly reduce $f_{\mathrm{cond}}$. Conversely, when $\lambda_{\mathrm{opa}} = 0$, the opacity remains unaffected by the Mahalanobis distance $D$, resulting in a constant $f_{\mathrm{cond}} = 1$ regardless of the viewing direction. When $\lambda_{\mathrm{opa}} = 0.35$, the Mahalanobis distance $D$ has a moderate influence on the opacity, providing a balance between sensitivity to the viewing direction and maintaining opacity over a wider range of directions.

## A.4 MORE ABLATION STUDY

Table 8: Ablation study on the Synthetic NeRF dataset.

| | | chair | drums | ficus | hotdog | lego | materials | mic | ship | avg |
|---|---|---|---|---|---|---|---|---|---|---|
| PSNR | No-SH | 34.87 | 26.10 | 31.91 | 37.50 | 34.78 | 30.13 | 35.49 | 30.34 | 32.64 |
| | No-$f_{\mathrm{cond}}$ | **35.91** | 26.25 | **34.09** | 37.70 | 35.18 | **30.73** | 34.82 | 30.27 | 33.12 |
| | No-$\lambda_{\mathrm{opa}}$ | 35.53 | 26.29 | 33.40 | 37.88 | 35.26 | 30.45 | 36.03 | **30.86** | 33.21 |
| | $\tau = 0.005$ | 35.22 | 26.36 | 33.07 | 37.52 | 35.03 | 30.60 | 35.57 | 30.49 | 32.98 |
| | $\lambda_{\mathrm{opa}} = 0.35$ | 35.51 | 26.37 | 33.39 | 37.82 | 35.22 | 30.71 | 35.98 | 30.60 | 33.20 |
| | learnable-$\lambda_{\mathrm{opa}}$ | 35.49 | **26.45** | 33.45 | **37.90** | **35.25** | 30.71 | **36.13** | 30.72 | **33.26** |
| # points | No-SH | **204,009** | 240,153 | **146,698** | **99,599** | **221,974** | 201,124 | 245,754 | 256,769 | **202,010** |
| | No-$f_{\mathrm{cond}}$ | 217,898 | 304,075 | 265,681 | 128,762 | 261,006 | 273,947 | 310,472 | 305,298 | 258,392 |
| | No-$\lambda_{\mathrm{opa}}$ | 204,335 | 242,526 | 188,295 | 101,039 | 223,815 | 210,190 | 265,266 | 271,282 | 213,344 |
| | $\tau = 0.005$ | 341,765 | 374,158 | 316,975 | 197,901 | 344,362 | 325,540 | 391,525 | 370,648 | 332,859 |
| | $\lambda_{\mathrm{opa}} = 0.35$ | 233,148 | 247,822 | 195,205 | 103,263 | 274,525 | 219,917 | 275,292 | 264,650 | 226,728 |
| | learnable-$\lambda_{\mathrm{opa}}$ | 223,227 | 250,267 | 197,741 | 102,451 | 233,807 | 222,209 | 272,052 | 270,163 | 221,482 |

Table 8 presents the results of our ablation studies on the Synthetic NeRF dataset (Mildenhall et al., 2020). Since this dataset lacks strong view-dependent effects, the influence of $\lambda_{\mathrm{opa}}$ is less significant compared to the 6DGS-PBR dataset (see Table 4). Compared to our model with $\lambda_{\mathrm{opa}} = 0.35$, setting *No-$f_{cond}$* (i.e., $\lambda_{\mathrm{opa}} = 0$) slightly degrades image quality and increases the number of Gaussian points. Conversely, when we set *No-$\lambda_{opa}$* (i.e., $\lambda_{\mathrm{opa}} = 1$), both the image quality and the number of Gaussians remain comparable. Notably, when we make $\lambda_{\mathrm{opa}}$ a learnable parameter, the model achieves improved image quality with a comparable number of Gaussian points. Furthermore, the *No-SH* setting results in degraded image quality. Setting the minimum opacity threshold to $\tau = 0.005$ (the default value in 3DGS) also degrades the performance and requires more Gaussian points.

Table 9: Training time (min) comparison of methods on the 6DGS-PBR dataset.

| | bunny-cloud | cloud | explosion | smoke | suzanne | ct-scan | avg |
|---|---|---|---|---|---|---|---|
| 3DGS | **29** | 38 | **19** | **40** | **41** | **11** | **30** |
| N-DG | 184 | *57* | 56 | 81 | 68 | 131 | 96 |
| **6DGS (Ours)** | 33 | ***35*** | 31 | 40 | 55 | 20 | 36 |

## A.5 TRAINING TIME COMPARISON

As shown in Table 9, our 6DGS demonstrates competitive training efficiency, requiring only 20% more time than 3DGS (36 vs 30 minutes) while being significantly faster than N-DG (96 minutes). Our training time could be further reduced by implementing Algorithm 1 in CUDA.

## A.6 MORE QUALITATIVE RESULTS

Figure 6 showcases additional qualitative results from the 6DGS-PBR dataset, highlighting the superior visual quality achieved by our 6DGS method compared to 3DGS and N-DG. These visualizations emphasize the ability of 6DGS to capture fine details and complex view-dependent effects while maintaining a high level of geometric fidelity.

Figures 7 and 8 provide qualitative results on three real-world datasets: Deep Blending (Hedman et al., 2021), Tanks & Temples (Knapitsch et al., 2017), and Shiny (Wizadwongsa et al., 2021). Across these datasets, 6DGS demonstrates comparable or superior visual quality to 3DGS. These results further validate the robustness and versatility of 6DGS in handling diverse and challenging scenarios, both synthetic and real-world.

Finally, we provide the Python implementation code for the 6DGS slicing algorithm in Listing 1.

```python
import torch

def slice_6dgs_to_3dgs(L, alpha, mu_p, mu_d, d, iteration,
        refine_iteratons, lambda_opa=0.35):
    # Compute the covariance matrix Sigma
    Sigma = torch.matmul(L, L.T)
    # Partition Sigma into blocks
    Sigma_p = Sigma[:, :3, :3]
    Sigma_pd = Sigma[:, :3, 3:]
    Sigma_d = Sigma[:, 3:, 3:]
    # Calculate the conditional mean, covariance, opacity
    Sigma_d_inv = torch.inverse(Sigma_d)
    Sigma_regr = torch.matmul(Sigma_pd, Sigma_d_inv)
    mu_cond = mu_p + torch.matmul(Sigma_regr, x.unsqueeze(-1)).squeeze()
    Sigma_cond = Sigma_p - torch.matmul(Sigma_regr, Sigma_pd.T)
    f_cond = torch.exp(-lambda_opa * torch.einsum('bi,bij,bj->b', x,
        Sigma_d_inv, x).unsqueeze(-1))
    alpha_cond = alpha * f_cond

    s, R = None, None
    if iteration in refine_iterations:
        # Perform SVD on the conditional covariance matrix
        U, S, V = torch.svd(Sigma_cond)
        # Extract the rotation matrix and scale
        R = U
        R[:, 2] *= torch.sign(torch.det(R).unsqueeze(-1))
        s = torch.sqrt(S)
    return mu_cond, Sigma_cond, alpha_cond, s, R
```

Listing 1: Python code for slicing 6DGS to conditional 3DGS

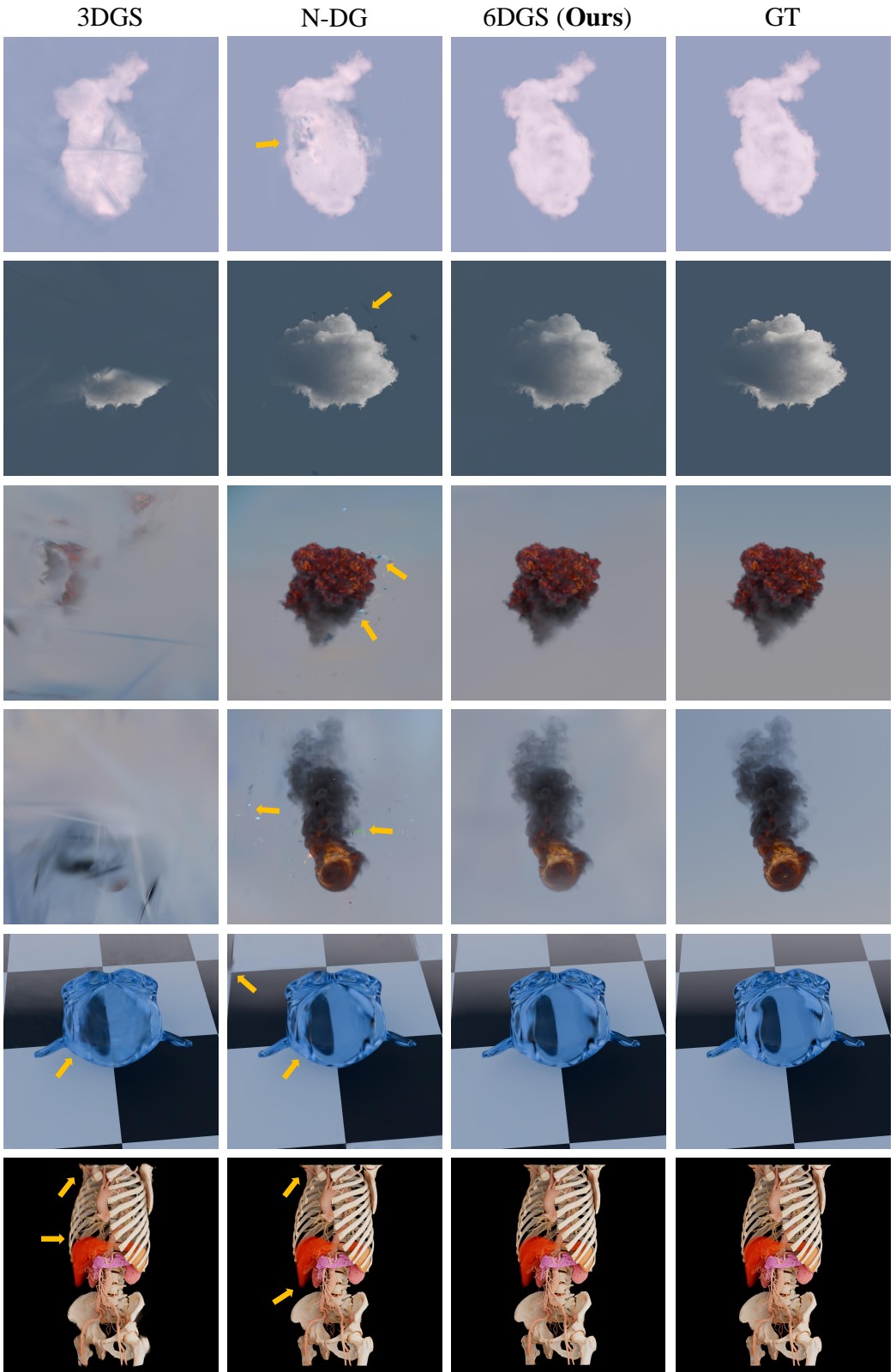

Figure 6: More qualitative comparison of methods on the 6DGS-PBR dataset (zoom in for details).

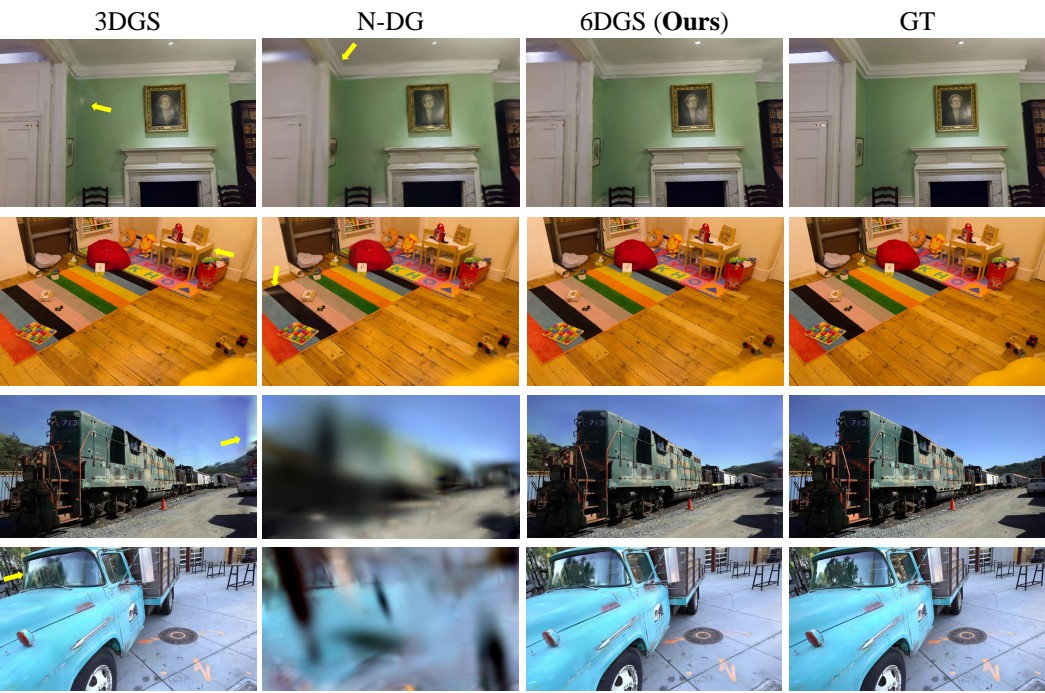

3DGS         N-DG         6DGS (**Ours**)         GT

Figure 7: Qualitative comparison of methods on the Deep Blending (Hedman et al., 2021) and Tanks & Temples (Knapitsch et al., 2017) datasets (zoom in for details).

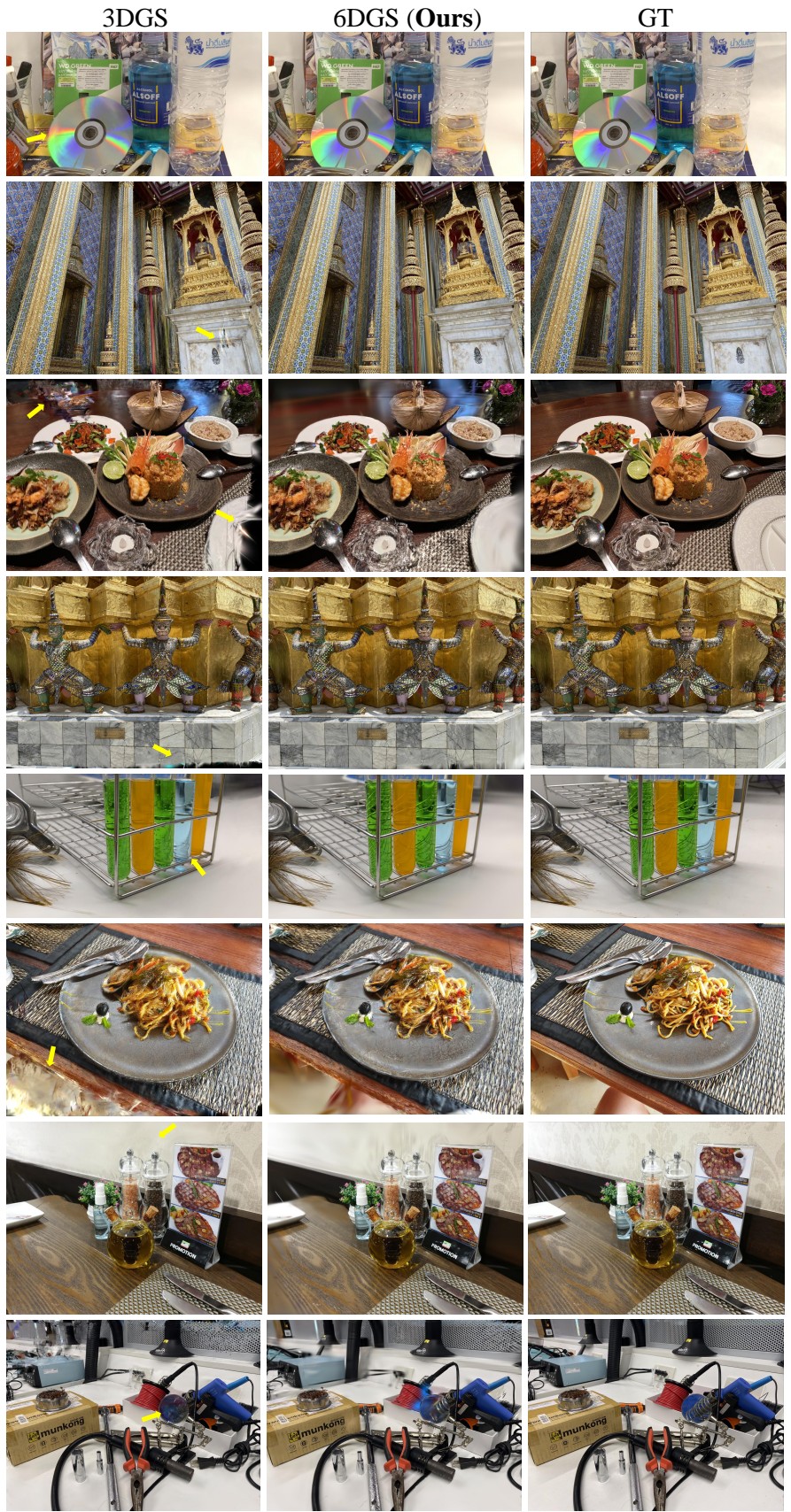

Figure 8: Qualitative comparison of methods on the Shiny dataset (Wizadwongsa et al., 2021).

