# OpenReview forum: "6DGS: Enhanced Direction-Aware Gaussian Splatting for Volumetric Rendering"
_ICLR.cc/2025/Conference — ICLR 2025 Poster_

### Official Review · Reviewer_cn4m · 2024-10-24

**Soundness:** 3
**Presentation:** 4
**Contribution:** 2
**Rating:** 6
**Confidence:** 4

**Summary:**

3DGS excels at synthesizing novel views of scenes, but it struggles with showing effects that depend on the viewer's perspective, like reflections, refractions, and scattering caused by complicated light transport. To solve this issue, the paper introduces a method called 6D-GS, which is based on N-dimensional GS. This method represents a scene as a 6D signal to capture both the spatial position and view direction at the same time. The paper enhances N-dimension GS by incorporating a view-dependent opacity function and an optimized density control. Extensive experiments on two synthetic datasets demonstrate the effectiveness of the proposed method.

**Strengths:**

- The paper is well-written and easy to follow. The details are clearly described, and I believe the results can be reproduced by an expert in this field.

- The method is simple and easy to implement. It allows slicing the 6D Gaussian given a view direction into a 3D Gaussian, which can then be directly fed into the existing 3D Gaussian rasterizer.

- Experiments show impressive results in the 6DGS-PBRT datasets, achieving 10dB over 3DGS and 1.5dB over N-D Gaussian. The results may interest research that aims to overfit high-dimension data.

**Weaknesses:**

The main weakness is the absence of supplemental video, which makes it difficult to access the novel view synthesis results.

From my understanding, allocating more capacity to model view dependency of a scene might hinder the geometry and novel view synthesis. For instance, early NeRF uses a relatively small MLP with fewer Fourier features to strike a balance.

Furthermore, the introduction of view-dependent opacity also complicates the results. For instance, while making the opacity view-dependent allows for easy fitting of high specular reflection, it is possible that when you view the specular spot from a slightly different angle, it becomes transparent. Since the PSNR metric may not provide a clear indication, it is recommended to include a video demonstration.

**Questions:**

I am interested in learning more about the potential applications of the algorithm. Can the algorithm be used for general radiance field reconstruction  instead of using synthetic data in dense view settings? If so, can you provide results from real-world datasets such as NeX?
If not, it would be helpful to understand the specific scenarios where algorithms can be applied and to discuss any limitations.

---

> ### Author Response · Authors · 2024-11-20
> **Response to Reviewer cn4m**
>
> Thank you for your detailed feedback and insightful suggestions. Below, we address your comments.
>
> ## W1. View-dependent Opacity and Supplemental Video
>
> We have validated the effectiveness of view-dependent opacity in our ablation studies (see rows labeled "No $\lambda_{opc}$" in Tables 4 and 5). The results clearly show the importance of this component in enhancing rendering quality for view-dependent scenes.
>
> Additionally, we have provided a supplemental video showcasing real-time rendering using a customized splatviz tool [1] on a standard laptop. This video demonstrates the practical capabilities of 6DGS in real-time applications and offers visual confirmation of its rendering quality for novel views.
>
> Reference:
> > [1] Barthel, Florian, et al. "Gaussian Splatting Decoder for 3D-aware Generative Adversarial Networks." arXiv preprint (2024).
>
> ## W2. Geometry and Novel View Synthesis
>
> We appreciate your concern about the trade-off between view dependency modeling and geometry reconstruction. However, our results indicate that 6DGS achieves a favorable balance:
> - Generalization: All our quantitative and qualitative evaluations are conducted on test set novel views, highlighting 6DGS's strong generalization capabilities.
> - Geometry Accuracy: Visualizations in Figure 3 (last three columns) illustrate that 6DGS produces more faithful geometry compared to 3DGS and N-DG.
> - Efficiency: By using significantly fewer Gaussian points than 3DGS (e.g., 67,490 vs. 115,111 points on average in the 6DGS-PBRT dataset), 6DGS avoids overfitting while improving both geometric and rendering accuracy.
>
> These results confirm that 6DGS effectively enhances view-dependent effects without compromising geometric fidelity or novel view synthesis quality.
>
> ## Q1. Real-world Scenes from NeX
>
> We have newly conducted extensive experiments on three real-world datasets: Deep Blending [1], Tanks&Temples [2], and Shiny from NeX [3]. Compared with 3DGS and N-DG:
> - **Deep Blending**: 6DGS achieves comparable quality (28.59 vs. 29.48 PSNR) to 3DGS with 33% fewer points.
> - **Tanks&Temples**: 6DGS outperforms 3DGS in quality (23.50 PSNR vs. 23.42 PSNR) while reducing the number of points by 22%.
> - **Shiny**: In scenes with strong view dependency (e.g., cd and lab), 6DGS achieves PSNR gains of +2.88 and +2.97, respectively, while utilizing only ~50% of the Gaussian points.
>
> These results demonstrate that 6DGS effectively generalizes to real-world scenarios, particularly excelling in scenes with strong view-dependent effects.
>
> In comparison to NeX, a method focused on implicit representations and specular effects, 6DGS achieves faster training times (48 minutes vs. >18 hours) with slightly lower average PSNR (23.01 vs. 26.45). This highlights 6DGS's trade-off between real-time performance and fidelity.
>
> | Dataset | Scene | 3DGS |  |  | N-DG |  |  | 6DGS (**ours**) |  |  |
> |---------|-------|------|------|----------|------|------|----------|--------------|------|----------|
> | | | PSNR | SSIM | # points | PSNR | SSIM | # points | PSNR | SSIM | # points |
> | Deep Blending | *drjohnson* | 29.22 | 0.898 | 3,276,989 | 26.31 | 0.828 | 494,494 | 28.12 | 0.883 | 2,074,490 |
> | | *playroom* | 29.74 | 0.900 | 2,332,830 | 27.74 | 0.866 | 258,999 | 29.06 | 0.892 | 1,685,626 |
> | | *avg* | 29.48 | 0.899 | 2,804,910 | 27.03 | 0.847 | 376,747 | 28.59 | 0.888 | 1,880,058 |
> ====================
> | Tanks&Temples | *train* | 21.75 | 0.803 | 1,100,525 | 13.26 | 0.440 | 996,826 | **21.95** | 0.787 | 839,408 |
> | | *truck* | 25.08 | 0.869 | 2,606,855 | 13.65 | 0.462 | 663,616 | 25.05 | 0.859 | 2,050,162 |
> | | *avg* | 23.42 | 0.836 | 1,853,690 | 13.46 | 0.451 | 830,221 | **23.50** | 0.823 | 1,444,785 |
> ====================
> | Shiny | *cd* | 25.51 | 0.843 | 1,128,098 | 17.98 | 0.530 | 120,536 | **28.39** | **0.895** | 596,789 |
> | | *crest* | 18.80 | 0.622 | 4,611,724 | 11.87 | 0.122 | 42,440 | **19.35** | **0.648** | 3,144,178 |
> | | *food* | 18.39 | 0.500 | 2,362,888 | 12.31 | 0.260 | 50,790 | 17.97 | 0.475 | 1,190,056 |
> | | *giants* | 24.24 | 0.844 | 2,341,337 | 14.72 | 0.300 | 39,117 | 24.15 | 0.826 | 1,635,315 |
> | | *lab* | 24.69 | 0.836 | 843,202 | 19.00 | 0.552 | 90,896 | **27.66** | **0.903** | 490,878 |
> | | *pasta* | 15.45 | 0.373 | 2,287,582 | 12.88 | 0.279 | 51,080 | 14.90 | 0.349 | 978,188 |
> | | *seasoning* | 26.25 | 0.823 | 1,085,732 | 14.06 | 0.425 | 41,665 | **26.36** | 0.811 | 524,848 |
> | | *tools* | 26.20 | 0.908 | 1,180,973 | 11.17 | 0.461 | 51,842 | 25.28 | 0.884 | 593,190 |
> | | *avg* | 22.44 | 0.719 | 1,980,192 | 14.25 | 0.366 | 61,046 | **23.01** | **0.724** | 1,144,180 |
>
> Reference:
> > [1] Hedman, Peter, et al. "Baking neural radiance fields for real-time view synthesis." ICCV 2021.
> >
> > [2] Knapitsch, Arno, et al. "Tanks and temples: Benchmarking large-scale scene reconstruction." ACM ToG 2017.
> >
> > [3] Wizadwongsa, Suttisak, et al. "Nex: Real-time view synthesis with neural basis expansion." CVPR 2021.
>
> CONTINUE ...

---

> > ### Author Response · Authors · 2024-11-20
> > **Response to Reviewer cn4m**
> >
> > ## Q2. Applications of 6DGS
> >
> > 6DGS is versatile and offers practical benefits for various applications:
> > 1. For view-dependent scenes:
> >     - Provides significant improvements in both rendering quality and speed
> >     - Achieves real-time performance while maintaining quality close to physically-based ray tracing
> >     - Suitable for medical visualization (e.g., interactive rendering of CT and MRI data).
> > 2. For scenes without strong view-dependency:
> >     - Maintain comparable quality to 3DGS while using substantially fewer Gaussian points
> >     - Suitable for applications like city-scale modeling due to enhanced efficiency.
> >     - Accelerates existing 3DGS workflows.
> >
> > The method's efficiency in point usage and strong view-dependent modeling makes it a practical choice for both specialized high-fidelity rendering and large-scale visualization tasks.

---

> > > ### Comment · Reviewer_cn4m · 2024-11-26
> > > **Lack video results on common datasets like NeRF-synthetic.**
> > >
> > > I appreciate your efforts, and most of my concerns have been addressed. However, I still have reservations regarding the novel view synthesis results. The supplemental video only includes a scene from a very dense capture and lacks comparisons. I would like to see results from a common benchmark dataset, such as NeRF-synthetic. More visualizations are well-come. Additionally, it would be helpful to include some baselines for comparison.

---

> ### Author Response · Authors · 2024-11-26
> **Response to Reviewer cn4m**
>
> Thank you for your thoughtful feedback. In response to your concern, we have included a new supplemental video featuring the `lego` scene from NeRF-Synthetic. Due to supplementary size limitations, we could only include one additional example. This visualization demonstrates that 6DGS achieves high-quality novel view synthesis. Notably, for novel views at the bottom of the `lego` model, 6DGS shows empty Gaussian points (**as the bottom is invisible in training images**), whereas 3DGS exhibits obvious artifacts.
>
> Additionally, we have added qualitative results in Section A6 of the Appendix, as shown in Figures 7 and 8. These figures provide visualizations from three real-world datasets: Deep Blending [1], Tanks & Temples [2], and Shiny [3]. Across these datasets, 6DGS demonstrates comparable or superior visual quality compared to baseline methods.
>
> We hope these additions address your concerns. Please feel free to let us know if you have further questions or suggestions. Thank you once again.

---

> > ### Comment · Reviewer_cn4m · 2024-11-28
> > **Comment by Reviewer cn4m**
> >
> > Thank you for providing the video for further clarification. However, I noticed that when viewed from the top of the LEGO base, it appears opaque, while from the bottom, it becomes transparent. This discrepancy may not be physically accurate. Although this inaccuracy might not pose a problem in dense view settings, it could affect the generalizability given insufficient viewpoints. Nevertheless, I believe the current method still offers advantages for certain tasks, particularly in an overfitting scenario so that I will maintain my positive scores.

---

> > > ### Author Response · Authors · 2024-11-28
> > > **Response to Reviewer cn4m**
> > >
> > > We would like to clarify that the bottom views of the `lego` scene are extrapolated, as the bottom is entirely occluded in the training set. In contrast, interpolated novel views, derived from perspectives present in the training data, are more representative and meaningful for evaluation, even in sparse view settings.
> > >
> > > We emphasize that 6DGS consistently outperforms 3DGS across multiple synthetic and real-world datasets, demonstrating superior generalizability. This is evident both quantitatively and qualitatively, as 6DGS achieves substantial improvements even in challenging scenarios. We greatly appreciate your recognition of our method’s strengths and its potential impact.

---

> ### Author Response · Authors · 2024-11-28
> **Response to Reviewer cn4m**
>
> Thank you for your insight and for maintaining a positive score.

---

### Official Review · Reviewer_zNRe · 2024-10-28

**Soundness:** 2
**Presentation:** 3
**Contribution:** 3
**Rating:** 6
**Confidence:** 4

**Summary:**

This paper aims to address the challenge of handling view-dependent effects in scenes, a common issue encountered in both NeRF and 3DGS. Inspired by N-DG, this paper further enhances the 6D spatial-angular representation to better model scene details, particularly in contexts exhibiting complex view-dependent effects. By leveraging additional directional information, the proposed method achieves an improved representation of opacity and color while utilizing fewer Gaussians, thereby enhancing rendering speed. In addition, this paper offers a theoretical analysis of the conditional Gaussian parameters derived from the 6DGS representation, revealing its physical significance and contributions to enhancing rendering quality.

**Strengths:**

- The method proposed in this paper enhances the modeling of opacity and color while reducing the scale of Gaussians. By slicing the 6DGS to conditional 3DGS, it further improves rendering speed.
- This paper offers a theoretical analysis of 6DGS that enhances rendering quality, aiding in the understanding of its contribution to modeling complex scenes, particularly the scenes with view-dependent effects.
- This paper clearly defines the problem to be addressed and proposes the 6DGS representation, which is beneficial for overcoming the limitations of previous representations in applications. Additionally, it provides some validation across multiple datasets.

**Weaknesses:**

- The experimental validation is insufficient. (1) The method proposed in this paper emphasizes its capability to model complex scenes exhibiting view-dependent effects; however, the qualitative experiments are limited to results derived from the 6DGS-PBRT dataset proposed in this paper. The authors should provide additional visualization results, such as those derived from the Synthetic NeRF dataset to strengthen the effectiveness of the proposed method. (2) The paper only presents results for two synthetic datasets, which does not adequately demonstrate the robustness and applicability of the proposed method in real-world scenarios.

- Some important analyses are missing. Due to the challenges of camera pose estimation in real-world scenes with view-dependent effects, and considering that camera pose influences Gaussian density control, the authors should consider conducting further analysis on this issue.

**Questions:**

Based on the aforementioned weaknesses, the following issues also require discussion:
- Given that N-DG demonstrates rendering results on real data, and as mentioned in line 62 of the paper, the proposed method can handle scenes with glossy surfaces and transparency, can the method achieve better modeling and rendering results when applied to the real-world dataset collected by NeX? As NeX proposes view-dependent pixel representation that can handle non-Lambertain surfaces and model fine details, the author should conduct comparative experiments with NeX.

[1] NeX: Real-time View Synthesis with Neural Basis Expansion, CVPR 2021.

- Since N-DG can be extended to a higher-dimensional representation, was the dimensional setting for this method in the comparative experiments set as 6D?

- As the method proposed in the paper is based on per-scene optimization, the authors should consider providing a comparison of training times across different scenes. This would help understand the contributions of the method better.

---

> ### Author Response · Authors · 2024-11-20
> **Response to Reviewer zNRe**
>
> Thank you for your thoughtful feedback and constructive suggestions. Below, we address each point in detail:
>
> ## W1. Synthetic NeRF and Real-world Scenes from NeX
>
> We would like to respectively clarify the reviewer `zNRe` that we have already included the Synthetic NeRF dataset, as shown in Table 3 and Table 5. Additionally, we have newly conducted extensive experiments on three real-world datasets: Deep Blending [1], Tanks&Temples [2], and Shiny from NeX [3]. Compared with 3DGS and N-DG:
> - **Deep Blending**: 6DGS achieves comparable quality (28.59 vs. 29.48 PSNR) to 3DGS with 33% fewer points.
> - **Tanks&Temples**: 6DGS outperforms 3DGS in quality (23.50 PSNR vs. 23.42 PSNR) while reducing the number of points by 22%.
> - **Shiny**: In scenes with strong view dependency (e.g., cd and lab), 6DGS achieves PSNR gains of +2.88 and +2.97, respectively, while utilizing only ~50% of the Gaussian points.
>
> These results demonstrate that 6DGS effectively generalizes to real-world scenarios, particularly excelling in scenes with strong view-dependent effects.
>
> In comparison to NeX, a method focused on implicit representations and specular effects, 6DGS achieves faster training times (48 minutes vs. >18 hours) with slightly lower average PSNR (23.01 vs. 26.45). This highlights 6DGS's trade-off between real-time performance and fidelity.
>
> | Dataset | Scene | 3DGS |  |  | N-DG |  |  | 6DGS (**ours**) |  |  |
> |---------|-------|------|------|----------|------|------|----------|--------------|------|----------|
> | | | PSNR | SSIM | # points | PSNR | SSIM | # points | PSNR | SSIM | # points |
> | Deep Blending | *drjohnson* | 29.22 | 0.898 | 3,276,989 | 26.31 | 0.828 | 494,494 | 28.12 | 0.883 | 2,074,490 |
> | | *playroom* | 29.74 | 0.900 | 2,332,830 | 27.74 | 0.866 | 258,999 | 29.06 | 0.892 | 1,685,626 |
> | | *avg* | 29.48 | 0.899 | 2,804,910 | 27.03 | 0.847 | 376,747 | 28.59 | 0.888 | 1,880,058 |
> ====================
> | Tanks&Temples | *train* | 21.75 | 0.803 | 1,100,525 | 13.26 | 0.440 | 996,826 | **21.95** | 0.787 | 839,408 |
> | | *truck* | 25.08 | 0.869 | 2,606,855 | 13.65 | 0.462 | 663,616 | 25.05 | 0.859 | 2,050,162 |
> | | *avg* | 23.42 | 0.836 | 1,853,690 | 13.46 | 0.451 | 830,221 | **23.50** | 0.823 | 1,444,785 |
> ====================
> | Shiny | *cd* | 25.51 | 0.843 | 1,128,098 | 17.98 | 0.530 | 120,536 | **28.39** | **0.895** | 596,789 |
> | | *crest* | 18.80 | 0.622 | 4,611,724 | 11.87 | 0.122 | 42,440 | **19.35** | **0.648** | 3,144,178 |
> | | *food* | 18.39 | 0.500 | 2,362,888 | 12.31 | 0.260 | 50,790 | 17.97 | 0.475 | 1,190,056 |
> | | *giants* | 24.24 | 0.844 | 2,341,337 | 14.72 | 0.300 | 39,117 | 24.15 | 0.826 | 1,635,315 |
> | | *lab* | 24.69 | 0.836 | 843,202 | 19.00 | 0.552 | 90,896 | **27.66** | **0.903** | 490,878 |
> | | *pasta* | 15.45 | 0.373 | 2,287,582 | 12.88 | 0.279 | 51,080 | 14.90 | 0.349 | 978,188 |
> | | *seasoning* | 26.25 | 0.823 | 1,085,732 | 14.06 | 0.425 | 41,665 | **26.36** | 0.811 | 524,848 |
> | | *tools* | 26.20 | 0.908 | 1,180,973 | 11.17 | 0.461 | 51,842 | 25.28 | 0.884 | 593,190 |
> | | *avg* | 22.44 | 0.719 | 1,980,192 | 14.25 | 0.366 | 61,046 | **23.01** | **0.724** | 1,144,180 |
>
> |   NeX Results    | *cd*  | *crest* | *food* | *giants* | *lab*  | *pasta* | *seasoning* | *tools* | *avg*  |
> |-------|-------|---------|--------|----------|--------|---------|-------------|---------|--------|
> | PSNR  | 31.43 | 21.23   | 23.68  | 26.00    | 30.43  | 22.07   | 28.60       | 28.16   | 26.45  |
>
> References:
> > [1] Hedman, Peter, et al. "Baking neural radiance fields for real-time view synthesis." Proceedings of the IEEE/CVF international conference on computer vision. 2021.
> >
> > [2] Knapitsch, Arno, et al. "Tanks and temples: Benchmarking large-scale scene reconstruction." ACM Transactions on Graphics (ToG) 36.4 (2017): 1-13.
> >
> > [3] Wizadwongsa, Suttisak, et al. "Nex: Real-time view synthesis with neural basis expansion." Proceedings of the IEEE/CVF Conference on Computer Vision and Pattern Recognition. 2021.
>
> ## W2. Challenges of Camera Pose
>
> We have evaluated 6DGS on three real-world datasets (Deep Blending, Tanks&Temples, and Shiny) where camera poses were estimated using COLMAP, presenting challenging real-world conditions. Our strong results on these datasets--particularly the improvements on scenes with strong view-dependent effects like 'cd' (+2.88 dB PSNR) and 'lab' (+2.97 dB PSNR)--demonstrate that 6DGS is robust to camera pose estimation uncertainties in real-world scenarios. This validation extends beyond synthetic datasets and confirms our method's effectiveness under practical deployment conditions.
>
> ## Q1. N-DG Settings
>
> Yes, we use the released code with default settings provided by N-DG, which uses 6D Gaussian representation for volumetric radiance fields. While N-DG also supports a 10D representation with additional features (albedo $rgb$ and roughness $r$), it is only used to demonstrate the application of Global Illumination.
>
> CONTINUE ...

---

> > ### Author Response · Authors · 2024-11-20
> > **Response to Reviewer zNRe**
> >
> > ## Q2. Training Time
> >
> > As shown in the table below, our 6DGS demonstrates competitive training efficiency, requiring only 20% more time than 3DGS (36 vs 30 minutes) while being significantly faster than N-DG (96 minutes). Training time could be further reduced by implementing Algorithm 1 in CUDA, as discussed with reviewer `3cPv`.
> >
> > |            | *bunny-cloud* | *cloud* | *explosion* | *smoke* | *suzanne* | *ct-scan* | *arg* |
> > |------------|---------------|---------|-------------|---------|-----------|-----------|-----------|
> > | 3DGS       | **29**       | *38*    | **19**      | **40**  | **41**    | **11**    | **30**         |
> > | N-DG       | 184          | *57*    | 56          | 81      | 68        | 131       |   96       |
> > | 6DGS (**Ours**)| 33           | ***35***| 31          | **40**  | 55        | 20        |  36    |
> >
> > We hope these clarifications and results address your concerns. Please let us know if additional details or experiments are needed.

---

> ### Comment · Reviewer_zNRe · 2024-11-26
> **Thanks for the rebuttal comments**
>
> I have read the other review comments and the rebuttal. The rebuttal has addressed most of my concerns; however, I still look forward to seeing the qualitative results in real-world scenarios. Therefore, I maintain my rating as marginally above the acceptance threshold.

---

> > ### Author Response · Authors · 2024-11-26
> > **Response to Reviewer zNRe**
> >
> > Thank you for your valuable feedback. To address your concern, we have included qualitative results in Section A6 of the Appendix, specifically in Figures 7 and 8. These figures present results on three real-world datasets: Deep Blending [1], Tanks & Temples [2], and Shiny [3].
> >
> > 6DGS demonstrates comparable or superior visual quality to baseline methods across these datasets. Notably, in the Shiny dataset, which includes challenging scenes like `crest`, `food`, `giants`, `pasta`, `seasoning`, and `tools`, our method consistently delivers high-quality results, even with fewer than 60 training images. In contrast, N-DG fails to converge under these constraints. For fairness, we used the officially released code and default settings for N-DG in all experiments. These results confirm the robustness and versatility of 6DGS across diverse and challenging scenarios, both synthetic and real-world.
> >
> > Please let us know if you have further questions. Thank you once again.

---

### Official Review · Reviewer_n9Tc · 2024-11-03

**Soundness:** 3
**Presentation:** 4
**Contribution:** 3
**Rating:** 8
**Confidence:** 3

**Summary:**

This paper proposes an extension of the existing 3D Gaussian Splatting technique to a six-dimensional Gaussian Splatting method. 6DGS combines 3D positional information with 3D view direction information using a six-dimensional Gaussian, enabling it to precisely capture view-dependent variations in color and transparency. To enhance computational efficiency, a restricted covariance method is introduced, which selectively incorporates only the necessary correlations, and spherical harmonics are employed to accurately model view-dependent color changes. Additionally, a new synthetic dataset called 6DGS-PBRT was constructed for performance evaluation, with experimental validation of view-dependent effects. This dataset, comprising synthetic scenes with various lighting and material properties, demonstrates that 6DGS can approximate complex visual effects effectively.

**Strengths:**

1. The fact that the proposed 6DGS is compatible with the existing 3D Gaussian Splatting (3DGS) framework and requires minimal modification appears to be a significant strength. This compatibility suggests that 6DGS can be effectively integrated into various existing datasets and application environments, potentially achieving impressive performance with minimal adjustments.

2. The theoretical explanations, figures, and experimental results are intuitively presented, making it easy to understand the contribution and performance of the proposed approach. This strong presentation quality greatly enhances readability.

3. The authors constructed a new 6DGS-PBRT dataset specifically to evaluate view-dependent effects, which seems to add considerable value. This dataset enables a more precise validation of 6DGS’s performance and appears to be an effective foundation for demonstrating its utility.

4. The approach provides important insights for the optimization of multi-dimensional Gaussian splatting techniques. By introducing restricted covariance modeling to refine view-dependent effects, this work offers valuable clues for future optimization of higher-dimensional Gaussian models.

**Weaknesses:**

1. Although experiments on synthetic datasets are meaningful, the lack of experiments on real-world captured datasets may limit the understanding of the model’s performance in real environments.

2. While the authors’ 6DGS-PBRT dataset seems well-suited for view-dependent effects, the limited variety of classes may fall short in representing a wider range of optical phenomena.

3. Additional explanation about the visual characteristics of each dataset would enhance clarity. For instance, clarifying that “cloud” captures absorption and scattering effects, or “glass” represents refraction effects, could make the paper more accessible. It also seems that adding reflective objects or other classes could provide a more comprehensive evaluation.

**Questions:**

1. Why does the proposed 6DGS outperform Ray Tracing-based Gaussian models on PBRT-rendered datasets? Intuitively, Ray Tracing-based models might be expected to perform better. Could it be due to incomplete implementation or insufficient optimization of the Ray Tracing-based model?

2. Although 6DGS is not Ray-Tracing-based, it appears to approximate the complex rendering effects and physical phenomena modeled by PBRT effectively. Could it also achieve high performance on more complex effects, such as subsurface scattering involving multiple scattering events?

3. While the synthetic dataset results already establish the value of this paper’s contributions, is there a specific reason or technical difficulty that prevented experiments on real-world datasets?

---

> ### Author Response · Authors · 2024-11-20
> **Response to Reviewer n9Tc**
>
> Thank you for highlighting the strengths of our work and the compatibility of 6DGS with existing frameworks. Below, we address your feedback and questions in detail.
>
> ## W1. Real-world Scene Evaluations
>
> To address the concern regarding real-world datasets, we have conducted additional experiments on three challenging datasets: Deep Blending [1], Tanks & Temples [2], and Shiny from NeX [3]. The results demonstrate that 6DGS generalizes well to real-world scenarios and excels in modeling strong view-dependent effects:
> - **Deep Blending**: 6DGS achieves comparable quality (28.59 vs. 29.48 PSNR) to 3DGS with 33% fewer points.
> - **Tanks&Temples**: 6DGS outperforms 3DGS in quality (23.50 PSNR vs. 23.42 PSNR) while reducing the number of points by 22%.
> - **Shiny**: In scenes with strong view dependency (e.g., cd and lab), 6DGS achieves PSNR gains of +2.88 and +2.97, respectively, while utilizing only ~50% of the Gaussian points.
>
> | Dataset | Scene | 3DGS |  |  | N-DG |  |  | 6DGS (**ours**) |  |  |
> |---------|-------|------|------|----------|------|------|----------|--------------|------|----------|
> | | | PSNR | SSIM | # points | PSNR | SSIM | # points | PSNR | SSIM | # points |
> | Deep Blending | *drjohnson* | 29.22 | 0.898 | 3,276,989 | 26.31 | 0.828 | 494,494 | 28.12 | 0.883 | 2,074,490 |
> | | *playroom* | 29.74 | 0.900 | 2,332,830 | 27.74 | 0.866 | 258,999 | 29.06 | 0.892 | 1,685,626 |
> | | *avg* | 29.48 | 0.899 | 2,804,910 | 27.03 | 0.847 | 376,747 | 28.59 | 0.888 | 1,880,058 |
> ====================
> | Tanks&Temples | *train* | 21.75 | 0.803 | 1,100,525 | 13.26 | 0.440 | 996,826 | **21.95** | 0.787 | 839,408 |
> | | *truck* | 25.08 | 0.869 | 2,606,855 | 13.65 | 0.462 | 663,616 | 25.05 | 0.859 | 2,050,162 |
> | | *avg* | 23.42 | 0.836 | 1,853,690 | 13.46 | 0.451 | 830,221 | **23.50** | 0.823 | 1,444,785 |
> ====================
> | Shiny | *cd* | 25.51 | 0.843 | 1,128,098 | 17.98 | 0.530 | 120,536 | **28.39** | **0.895** | 596,789 |
> | | *crest* | 18.80 | 0.622 | 4,611,724 | 11.87 | 0.122 | 42,440 | **19.35** | **0.648** | 3,144,178 |
> | | *food* | 18.39 | 0.500 | 2,362,888 | 12.31 | 0.260 | 50,790 | 17.97 | 0.475 | 1,190,056 |
> | | *giants* | 24.24 | 0.844 | 2,341,337 | 14.72 | 0.300 | 39,117 | 24.15 | 0.826 | 1,635,315 |
> | | *lab* | 24.69 | 0.836 | 843,202 | 19.00 | 0.552 | 90,896 | **27.66** | **0.903** | 490,878 |
> | | *pasta* | 15.45 | 0.373 | 2,287,582 | 12.88 | 0.279 | 51,080 | 14.90 | 0.349 | 978,188 |
> | | *seasoning* | 26.25 | 0.823 | 1,085,732 | 14.06 | 0.425 | 41,665 | **26.36** | 0.811 | 524,848 |
> | | *tools* | 26.20 | 0.908 | 1,180,973 | 11.17 | 0.461 | 51,842 | 25.28 | 0.884 | 593,190 |
> | | *avg* | 22.44 | 0.719 | 1,980,192 | 14.25 | 0.366 | 61,046 | **23.01** | **0.724** | 1,144,180 |
>
> References:
> > [1] Hedman, Peter, et al. "Baking neural radiance fields for real-time view synthesis." Proceedings of the IEEE/CVF international conference on computer vision. 2021.
> >
> > [2] Knapitsch, Arno, et al. "Tanks and temples: Benchmarking large-scale scene reconstruction." ACM Transactions on Graphics (ToG) 36.4 (2017): 1-13.
> >
> > [3] Wizadwongsa, Suttisak, et al. "Nex: Real-time view synthesis with neural basis expansion." Proceedings of the IEEE/CVF Conference on Computer Vision and Pattern Recognition. 2021.
>
> ## W2. Visual Characteristics and Dataset Variety
>
> Our 6DGS-PBRT dataset is designed to test a diverse range of optical phenomena using six carefully crafted scenes rendered with the PBRT engine "Cycle." These scenes address various challenging light transport effects:
> - **Volumetric Effects:**
>     - Scenes: *cloud-bunny*, *cloud*, *smoke*, *explosion*, *ct-scan*
>     - Effects: Complex volumetric absorption and scattering
>     - Goal: Test the model’s capability to handle varying densities and light transport through participating media
>
> - **Surface Effects:**
>     - Scenes: *suzanne*
>     - Effects: Translucence and refraction through glass BSDF material, along with surface scattering
>     - Goal: Evaluate the model’s ability to manage intricate light-material interactions
>
> To further validate the versatility of our method, we added a new dragon scene that combines subsurface scattering (SSS) and surface scattering effects. As shown in the table below, 6DGS achieved the best results with the fewest Gaussian points. Visualizations of the dragon scene are provided in the Supplementary Material. These additional evaluations demonstrate the adaptability of 6DGS to diverse scenarios involving both volumetric and surface-based light transport phenomena.
>
>
> | `SSS Dragon `| PSNR | SSIM | # points | FPS | Train (min) |
> |------------|------|------|----------|-----|--------|
> | 3DGS | 26.57 | 0.813 | 269,250 | 104.6 | **24** |
> | N-DG | 33.19 | 0.936 | 196,645 | 86.0 | 50 |
> | 6DGS (**Ours**) | **35.00** | **0.937** | **128,748** | **111.5** | 33 |
> | 6DGS-Flash (**Ours**) | - | - | - | **324.3** | - |
>
> CONTINUE ...

---

> > ### Author Response · Authors · 2024-11-20
> > **Response to Reviewer n9Tc**
> >
> > ## Q1. Gaussian-based Ray Tracing Comparison
> >
> > We would like to clarify a potential misunderstanding regarding comparisons with Gaussian-based ray tracing methods. Our 6DGS-PBRT dataset serves as a ground truth for training and testing, generated through physically-based ray tracing. We do not directly compare 6DGS with Gaussian-based ray tracing methods because these methods face inherent trade-offs, including slower rendering speeds and potentially lower image quality.
> >
> > As highlighted in Related Work, 6DGS takes a different approach, efficiently slicing 6D Gaussians into conditional 3D Gaussians for fast rasterization. This approach allows 6DGS to approximate high-fidelity ray-traced results while achieving real-time performance. Therefore, we focused on comparative evaluations against other rasterization-based Gaussian solutions (3DGS, N-DG), which are more relevant baselines for our method.
> >
> > It is worth noting that none of the recent Gaussian-based ray-tracing methods had publicly available implementations at the time of our submission. Since then, Blanc et al. (2024) has shared their code. However, we have faced challenges running their implementation on new scenes due to compatibility issues. Future work may explore a more in-depth comparison once these challenges are resolved.
> >
> > We hope these clarifications address your concerns and appreciate your valuable feedback. Please let us know if further details are required.

---

> > > ### Comment · Reviewer_n9Tc · 2024-12-02
> > >
> > > Dear Author,
> > >
> > > Your methodology is intuitive, and the explanation is clear. Furthermore, the experiments have demonstrated its effectiveness. The additional experiments have reinforced my confidence in my evaluation. Therefore, I stand by my assessment.

---

> > > > ### Author Response · Authors · 2024-12-02
> > > > **Response to Reviewer n9Tc**
> > > >
> > > > Thank you for your positive feedback and kind words regarding our methodology, explanation, and experimental results. We are delighted that the additional experiments reinforced your confidence in our work and appreciate your thoughtful evaluation.

---

### Official Review · Reviewer_3cPv · 2024-11-04

**Soundness:** 4
**Presentation:** 3
**Contribution:** 4
**Rating:** 8
**Confidence:** 4

**Summary:**

This paper presents 6D Gaussian Splatting(6DGS), an advanced method for volumetric rendering that enhances traditional 3D Gaussian splatting (3DGS) by incorporating view-dependent information through a 6D spatial-angular Gaussian representation.

The authors address key limitations in 3DGS, such as handling specular reflections and anisotropic effects, by leveraging a 6D Gaussian model that adapts both color and opacity based on viewing angles. Experimental results demonstrate significant improvements in both image quality (up to a 15.73 dB PSNR increase) and rendering efficiency, achieving real-time speeds on complex datasets.

**Strengths:**

The paper presents a novel 6D Gaussian Splatting (6DGS) method, which effectively models view-dependent effects through spatial and directional information.

By improving color and opacity handling, 6DGS produces high-quality, realistic renderings, particularly useful in scenes with complex reflections and lighting. Its compatibility with the 3DGS framework enhances its practical relevance, allowing integration with minimal adjustments. The experiments are well-designed and validate the method's efficiency, achieving significant reductions in Gaussian points without compromising image quality.

**Weaknesses:**

The method is primarily tested on synthetic datasets, lacking real-world scene evaluations that would confirm its robustness in practical applications. The new parameter λopa adds complexity, as it requires careful tuning. Additionally, while efficient for static scenes, the model’s performance on dynamic, interactive scenes remains untested, limiting its demonstrated versatility. Finally, the initialization phase can be computationally intensive, posing a potential challenge for scaling to large or complex scenes.

**Questions:**

- Could the authors provide more insights into how the choice of λopa influences performance across diverse datasets?

- How would the model adapt or perform in dynamic scenes with frequent view changes?

- Would implementing 6DGS directly in CUDA offer significant speed improvements for real-time applications, and if so, are there plans for such an implementation?

**Details Of Ethics Concerns:**

No specific ethical concerns identified.

---

> ### Author Response · Authors · 2024-11-20
> **Response to Reviewer 3cPv**
>
> Thank you for your insightful feedback and constructive comments. Below, we address each point in detail:
>
> ## W1. Real-world Scene Evaluations
>
> We have newly conducted extensive experiments on three real-world datasets: Deep Blending [1], Tanks&Temples [2], and Shiny from NeX [3], as shown in Table 5 in the Appendix. Our key observations include:
> - **Deep Blending**: 6DGS achieves comparable quality (28.59 vs. 29.48 PSNR) to 3DGS with 33% fewer points.
> - **Tanks&Temples**: 6DGS outperforms 3DGS in quality (23.50 PSNR vs. 23.42 PSNR) while reducing the number of points by 22%.
> - **Shiny**: In scenes with strong view dependency (e.g., cd and lab), 6DGS achieves PSNR gains of +2.88 and +2.97, respectively, while utilizing only ~50% of the Gaussian points.
>
> These results demonstrate the robustness of 6DGS in generalizing to real-world settings and its ability to excel in modeling strong view-dependent effects.
>
> | Dataset | Scene | 3DGS |  |  | N-DG |  |  | 6DGS (**ours**) |  |  |
> |---------|-------|------|------|----------|------|------|----------|--------------|------|----------|
> | | | PSNR | SSIM | # points | PSNR | SSIM | # points | PSNR | SSIM | # points |
> | Deep Blending | *drjohnson* | 29.22 | 0.898 | 3,276,989 | 26.31 | 0.828 | 494,494 | 28.12 | 0.883 | 2,074,490 |
> | | *playroom* | 29.74 | 0.900 | 2,332,830 | 27.74 | 0.866 | 258,999 | 29.06 | 0.892 | 1,685,626 |
> | | *avg* | 29.48 | 0.899 | 2,804,910 | 27.03 | 0.847 | 376,747 | 28.59 | 0.888 | 1,880,058 |
> ====================
> | Tanks&Temples | *train* | 21.75 | 0.803 | 1,100,525 | 13.26 | 0.440 | 996,826 | **21.95** | 0.787 | 839,408 |
> | | *truck* | 25.08 | 0.869 | 2,606,855 | 13.65 | 0.462 | 663,616 | 25.05 | 0.859 | 2,050,162 |
> | | *avg* | 23.42 | 0.836 | 1,853,690 | 13.46 | 0.451 | 830,221 | **23.50** | 0.823 | 1,444,785 |
> ====================
> | Shiny | *cd* | 25.51 | 0.843 | 1,128,098 | 17.98 | 0.530 | 120,536 | **28.39** | **0.895** | 596,789 |
> | | *crest* | 18.80 | 0.622 | 4,611,724 | 11.87 | 0.122 | 42,440 | **19.35** | **0.648** | 3,144,178 |
> | | *food* | 18.39 | 0.500 | 2,362,888 | 12.31 | 0.260 | 50,790 | 17.97 | 0.475 | 1,190,056 |
> | | *giants* | 24.24 | 0.844 | 2,341,337 | 14.72 | 0.300 | 39,117 | 24.15 | 0.826 | 1,635,315 |
> | | *lab* | 24.69 | 0.836 | 843,202 | 19.00 | 0.552 | 90,896 | **27.66** | **0.903** | 490,878 |
> | | *pasta* | 15.45 | 0.373 | 2,287,582 | 12.88 | 0.279 | 51,080 | 14.90 | 0.349 | 978,188 |
> | | *seasoning* | 26.25 | 0.823 | 1,085,732 | 14.06 | 0.425 | 41,665 | **26.36** | 0.811 | 524,848 |
> | | *tools* | 26.20 | 0.908 | 1,180,973 | 11.17 | 0.461 | 51,842 | 25.28 | 0.884 | 593,190 |
> | | *avg* | 22.44 | 0.719 | 1,980,192 | 14.25 | 0.366 | 61,046 | **23.01** | **0.724** | 1,144,180 |
>
> References:
> > [1] Hedman, Peter, et al. "Baking neural radiance fields for real-time view synthesis." ICCV 2021.
> >
> > [2] Knapitsch, Arno, et al. "Tanks and temples: Benchmarking large-scale scene reconstruction." ACM ToG 36.4 (2017): 1-13.
> >
> > [3] Wizadwongsa, Suttisak, et al. "Nex: Real-time view synthesis with neural basis expansion." CVPR 2021.
>
> ## Q1. $\lambda_{opa}$ Parameter Analysis
>
> We agree that the parameter $\lambda_{opa}$ is a critical component in 6DGS. As shown in Table 4 (ablation study), even when naively setting $\lambda_{opa}=1$, 6DGS significantly outperforms 3DGS, achieving +8.52dB PSNR (36.05 vs 27.53) while using 48.7% fewer Gaussian points. We introduce $0 < \lambda_{opa} < 1$ as a hyper-parameter to fine-tune view-dependency effects, where higher $\lambda_{opa}$ leads to stronger view-dependency with fewer Gaussian points (Appendix Figure 4). With our default setting of $\lambda_{opa}=0.35$, we further improve PSNR to 37.49 (+1.44dB) with a slight increase in points (66,803), still 42% fewer than 3DGS. Making $\lambda_{opa}$ learnable achieves the best PSNR (37.61) while maintaining an efficient point count (67,492). Importantly, our method achieves state-of-the-art performance across multiple datasets using the default $\lambda_{opa}=0.35$ without dataset-specific tuning.
>
> | Scene | $\lambda_{opa}=1$ |  | $\lambda_{opa}=0.35$ |  | $\lambda_{opa}$ learnable |  | 3DGS |  |
> |-------|---------|------------|------------|------------|----------------|------------|-------|------------|
> |  | *PSNR* | *# points* | *PSNR* | *# points* | *PSNR* | *# points* | *PSNR* | *# points* |
> |*bunny-cloud* | 39.12 | 4,860 | 40.47 | 6,830 | **41.57** | 6,660 | 30.75 | 21,074 |
> |*cloud* | 39.86 | 11,736 | **40.73** | 12,454 | 40.42 | 12,657 | 29.70 | 58,233 |
> |*explosion* | 40.55 | 13,582 | **42.69** | 17,051 | 42.48 | 17,133 | 26.75 | 51,140 |
> |*smoke* | 36.88 | 10,540 | 40.45 | 10,570 | **40.61** | 10,762 | 28.55 | 60,533 |
> |*suzanne* | 26.92 | 158,041 | **27.15** | 172,373 | 27.03 | 174,756 | 23.70 | 270,001 |
> |*ct-scan* | 32.99 | 154,997 | 33.42 | 181,539 | **33.56** | 182,981 | 25.71 | 229,683 |
> |*avg* | 36.05 | 58,959 | 37.49 | 66,803 | **37.61** | 67,492 | 27.53 | 115,111 |
>
> CONTINUE ...

---

> ### Author Response · Authors · 2024-11-20
> **Response to Reviewer 3cPv**
>
> ## Q2. Dynamic Scenes
>
> We thank the reviewer for raising this question, which we have been considering with too. Our 6DGS improves the view-dependency modeling ability of 3DGS and can be naturally extended to dynamic scenes in two ways. First, we can directly integrate our directional component into existing 3DGS-based dynamic methods [4, 5]. Second, we can extend 6DGS with an additional temporal dimension to model a 7D spatial-angular-temporal Gaussian:
>
> $$
> \begin{bmatrix}
> X_p \newline
> X_t \newline
> X_d
> \end{bmatrix} \sim
> \mathcal{N}(
> \begin{bmatrix}
> \mu_p \newline
> \mu_t \newline
> \mu_d
> \end{bmatrix},
> \begin{bmatrix}
> \Sigma_{pp} & \Sigma_{pt} & \Sigma_{pd} \newline
> \Sigma_{tp} & \sigma_{tt} & \Sigma_{td} \newline
> \Sigma_{dp} & \Sigma_{dt} & \Sigma_{dd}
> \end{bmatrix}
> )
> $$
>
> where $\mu_p \in \mathbb{R}^3$, $\mu_d \in \mathbb{R}^3$, and $\mu_t \in \mathbb{R}$ represent the means in position, direction, and temporal space respectively, and $\Sigma \in \mathbb{R}^{7\times7}$ captures the full covariance. While extending 6DGS to dynamic scenes is an exciting future direction, it is beyond the scope of our current work, which focuses on demonstrating the effectiveness of 6DGS for static scenes.
>
> References:
> >[4] Wu, Guanjun, et al. "4d gaussian splatting for real-time dynamic scene rendering." Proceedings of the IEEE/CVF Conference on Computer Vision and Pattern Recognition. 2024.
> >
> >[5] Yang, Ziyi, et al. "Deformable 3d gaussians for high-fidelity monocular dynamic scene reconstruction." Proceedings of the IEEE/CVF Conference on Computer Vision and Pattern Recognition. 2024.
>
> ## Q3. Initialization Phase
>
> We respectfully disagree with the concern about computational complexity in initialization. In fact, our initialization is computationally simpler than 3DGS:
>
> - Position and opacity: Uses the same initialization method as 3DGS
> - Scale initialization: Simply sets diagonal elements in the covariance matrix to 0.1, while 3DGS requires computing point distances
> - Rotation initialization: Sets off-diagonal elements to zero, similar to 3DGS's zero rotation
>
> Therefore, 6DGS initialization has similar or lower computational cost compared to 3DGS.
>
> ## Q4. CUDA Implementation
>
> We acknowledge that implementing Algorithm 1 in CUDA could further enhance speed. However, the performance gain would not be significant since Gaussian rasterization remains the primary computational bottleneck. Instead, we focused on accelerating the rendering pipeline by adopting FlashGS, resulting in 6DGS-Flash which significantly improves performance from 154.5 FPS to 326.3 FPS (2.1× speedup) as shown in Table 2. We are also considering implementing Algorithm 1 in CUDA to further optimize our training and rendering pipeline.
>
> We hope these clarifications address your concerns and appreciate your valuable feedback.

---

### Author Response · Authors · 2024-11-20
**Summary Response to All Reviewers**

We are very grateful to the reviewers `3cPv`, `n9Tc`, `zNRe`, and `cn4m` for their constructive feedback, as well as the recognition of our paper's strengths, such as its novelty in enhancing direction-aware gaussian splatting, significant performance improvements, and compatibility with existing frameworks.

In this global response, we highlight the shared strengths of our work as noted by the reviewers and address the main shared concern. Individual questions are addressed in detail in our responses to each reviewer, and we look forward to continued discussion.

**Shared Strengths**

All reviewers commended the following strengths of our work:
- *Technical Contribution*: Strong technical innovation with significant performance gains (up to +15.73 dB PSNR with 66.5% fewer Gaussian points).
- *Framework Compatibility*: Seamless integration with the existing 3DGS framework, enabling easy adoption and practical usability.
- *Clarity and Analysis*: Well-structured presentation, clear theoretical explanations, and detailed experimental validation.

**Shared Questions**

A primary concern shared by reviewers `3cPv`, `n9Tc`, `zNRe`, and `cn4m` was the lack of real-world dataset evaluations in the original submission. We acknowledge this valid point and have addressed it by conducting new experiments on three real-world datasets. Key findings are as follows:
- *Deep Blending*: 6DGS achieves comparable quality to 3DGS (28.59 vs. 29.48 PSNR) while using 33% fewer Gaussian points.
- *Tanks & Temples*: 6DGS outperforms 3DGS in quality (23.50 vs. 23.42 PSNR) while using 22% fewer points.
- *Shiny*: In challenging scenes with strong view dependency (cd and lab), 6DGS achieves PSNR gains (+2.88 and +2.97, respectively) while using only ~50% of the Gaussian points.

These results demonstrate that 6DGS generalizes effectively to real-world scenarios and excels in scenes with view-dependent effects.

**New Results and Submissions Edits**

In response to the reviewers' feedback, we have submitted an updated version of our manuscript and supplementary material, incorporating the following changes:
- *New Experiments*: Comprehensive evaluations on real-world datasets (Deep Blending, Tanks & Temples, and Shiny) and a subsurface scattering (SSS) dragon scene.
- *Expanded Supplementary Material*: Includes new video demonstrations showcasing real-time rendering capabilities, as well as visualizations.
- *Clarifications and Analyses*: Added detailed explanations to address reviewer concerns, such as view-dependent opacity and training efficiency.

---

### Comment · Area_Chair_FRHJ · 2024-11-25
**Please read the rebuttal and reply**

Dear Reviewers,

Thanks again for serving for ICLR, the discussion period between authors and reviewers is approaching (November 27 at 11:59pm AoE), please read the rebuttal and ask questions if you have any. Your timely response is important and highly appreciated.

Thanks,

AC

---

### Meta-Review · Area_Chair_FRHJ · 2024-12-17

**Metareview:**

This paper proposes a novel 3D Gaussians representation that can model view-dependent effects, and thus achieve better results on transparent or glossy materials. The proposed method achieves better performance compared to native 3DGS as well as N-D GS, the fact that it can be compatible with existing 3DGS pipeline also makes it more practical. The reviewers raised weaknesses such as missing experiments (e.g., testing on real-world data), robustness of newly introduced parameter, differentiation with Ray Tracing-based Gaussian models, missing supplementary videos, most of the issues are addressed during the rebuttal and reviewers all agree to accept this paper.

**Additional Comments On Reviewer Discussion:**

During rebuttal, the reviewers raised the following points:
- missing experiments (e.g., testing on real-world data)
- robustness of newly introduced parameter
- differentiation with Ray Tracing-based Gaussian models,
- missing supplementary videos

Most of these issues are addressed during the rebuttal and reviewers all agree to accept this paper. The authors are encouraged to include additional examples, especially more examples in the supplementary video, which will help the readers to better appreciate the performance of the proposed method.

---

### Decision · Program_Chairs · 2025-01-22

Accept (Poster)